# Fn14 promotes myoblast fusion during regenerative myogenesis

Meiricris Tomaz da Silva[1] , Aniket S Joshi[1], Micah B Castillo[2], Tatiana E Koike[1], Anirban Roy[1] , Preethi H Gunaratne[2], Ashok Kumar[1]

**Skeletal muscle regeneration involves coordinated activation of an array of signaling pathways. Fibroblast growth factor–inducible 14 (Fn14) is a bona fide receptor for the TWEAK cytokine. Levels of Fn14 are increased in the skeletal muscle of mice after injury. However, the cell-autonomous role of Fn14 in muscle regeneration remains unknown. Here, we demonstrate that global deletion of the Fn14 receptor in mice attenuates muscle regeneration. Conditional ablation of Fn14 in myoblasts but not in differentiated myofibers of mice inhibits skeletal muscle regeneration. Fn14 promotes myoblast fusion without affecting the levels of myogenic regulatory factors in the regenerating muscle. Fn14 deletion in myoblasts hastens initial differentiation but impairs their fusion. The overexpression of Fn14 in myoblasts results in the formation of myotubes having an increased diameter after induction of differentiation. Ablation of Fn14 also reduces the levels of various components of canonical Wnt and calcium signaling both in vitro and in vivo. Forced activation of Wnt signaling rescues fusion defects in Fn14-deficient myoblast cultures. Collectively, our results demonstrate that Fn14-mediated signaling positively regulates myoblast fusion and skeletal muscle regeneration.**

## Introduction

The skeletal muscle is composed of elongated syncytia (or myofibers) that are formed by the fusion of hundreds of mononucleated myoblasts during embryonic development. The skeletal muscle has a remarkable capability to regenerate, which is attributed to the presence of muscle stem cells, named satellite cells. These cells are localized between sarcolemma and basal lamina in a relatively quiescent state (Yin et al, 2013; Tidball, 2017; Relaix et al, 2021). After injury, satellite cells undergo several rounds of proliferation followed by their differentiation into myoblasts. Eventually, myoblasts fuse with each other or with injured myofibers to accomplish muscle repair

(Yin et al, 2013). Myoblast fusion is not only crucial for myofiber regeneration but also essential for myofiber growth in response to resistance exercise (Rochlin et al, 2010; Sampath et al, 2018). Accumulating evidence suggests that several transmembrane, cytoskeletal, and intracellular proteins mediate the fusion of myoblasts in vertebrates (Hindi et al, 2013; Millay et al, 2013; Bi et al, 2017; Leikina et al, 2018; Hindi & Millay, 2022). In addition, myoblast fusion involves activation of multiple signaling cascades, such as canonical Wnt, non-canonical NF-κB, calcineurin–NFATc2, Rho GTPases, integrin–focal adhesion kinase, and MAPKs (Hindi et al, 2013; Hindi & Millay, 2022; Sinha et al, 2022). However, molecular underpinning regulating myoblast fusion and skeletal muscle regeneration remains less understood.

Innate immune response and inflammatory cytokines play important roles in the regulation of skeletal muscle regeneration and growth (Tidball, 2017; Yang & Hu, 2018). TNF-like weak inducer of apoptosis (TWEAK, gene name: *TNFSF12*), a member of the TNF superfamily, is a multifunctional cytokine that functions through binding to the fibroblast growth factor–inducible 14 (Fn14, gene name: *TNFRSF12A*) receptor (Chicheportiche et al, 1997; Meighan-Mantha et al, 1999; Wiley et al, 2001; Winkles et al, 2007). TWEAK is constitutively expressed by cells of the innate immune system and some non-hematopoietic cell types (Zheng & Burkly, 2008; Burkly, 2014; Tajrishi et al, 2014). In contrast, the Fn14 receptor is expressed at a relatively low level in uninjured healthy tissues. The activity of the TWEAK-Fn14 system is enhanced because of the local expression of Fn14 after injury in many tissues, including skeletal muscles, arteries, and liver and in various disease states, such as multiple sclerosis, rheumatoid arthritis, systemic lupus erythematosus, and cancer leading to activation of MAPKs and canonical and non-canonical NF-κB signaling (Winkles, 2008; Mittal et al, 2010b; Burkly, 2014; Tajrishi et al, 2014; Poveda et al, 2021). The TWEAK-Fn14 axis regulates inflammation, angiogenesis, and cell survival, proliferation, migration, and differentiation in different organs (Chicheportiche et al, 1997; Meighan-Mantha et al, 1999; Wiley et al, 2001; Winkles et al, 2007; Burkly, 2014). Intriguingly, the overexpression of Fn14, which occurs in some types of cancers, can also activate downstream signaling pathways without requiring

---

[1]Department of Pharmacological and Pharmaceutical Sciences, University of Houston College of Pharmacy, Houston, TX, USA   [2]Department of Biology and Biochemistry, University of Houston, Houston, TX, USA

Correspondence: akumar43@Central.UH.EDU

stimulation by the TWEAK cytokine (Winkles, 2008; Burkly, 2014; Tajrishi et al, 2014).

Previous studies have demonstrated that the Fn14 receptor is expressed on the surface of muscle progenitor cells (Girgenrath et al, 2006; Dogra et al, 2007b). Addition of recombinant TWEAK protein induces the proliferation of cultured myoblasts but inhibits their differentiation into myotubes (Dogra et al, 2006, 2007b; Girgenrath et al, 2006). Moreover, the transgenic overexpression of TWEAK in the skeletal muscle of mice inhibits regeneration, whereas TWEAK-KO mice demonstrate improvements in myofiber regeneration and growth after injury (Mittal et al, 2010b). Consistent with its role in inhibiting myogenic differentiation, several other studies have demonstrated that the activity of the TWEAK-Fn14 axis is enhanced in degenerative muscle diseases, such as burn injury (Merritt et al, 2013), myotonic dystrophy (Yadava et al, 2015, 2016), amyotrophic lateral sclerosis (Bowerman et al, 2015), inclusion-body myositis (Morosetti et al, 2012), and spinal cord injury–associated muscle atrophy and fibrosis (Yarar-Fisher et al, 2016). Intriguingly, in contrast to TWEAK-KO mice, skeletal muscle regeneration is reduced in whole-body Fn14-KO mice. It was reported that the levels of monocyte chemotactic protein 3 are reduced in the injured muscle of Fn14-KO mice, which may lead to the diminished innate immune response required for the removal of tissue debris after muscle damage (Girgenrath et al, 2006). However, it remained unknown how the loss of Fn14 affects various parameters of muscle regeneration, such as the expression of myogenic regulatory factors (MRFs) and abundance of muscle progenitor cells and their fusion with injured myofibers. Furthermore, the cell-autonomous role of Fn14 in skeletal muscle regeneration has not yet been investigated. Specifically, it remained unknown whether Fn14-mediated signaling in muscle progenitor cells or differentiated myofibers influences skeletal muscle regeneration after injury.

In this study, using whole-body and conditional Fn14-KO mice, we have investigated the role and mechanisms of action of Fn14 in skeletal muscle regeneration. Our results demonstrate that deletion of Fn14 in myoblasts but not in myofibers inhibits skeletal muscle regeneration in adult mice. Although the loss of Fn14 does not inhibit myogenic differentiation, it reduces myoblast fusion both in vivo and in vitro. The overexpression of Fn14 in cultured myoblasts leads to the formation of myotubes having an increased diameter. Our results also demonstrate that Fn14 stimulates myoblast fusion through activation of canonical Wnt signaling.

# Results

## Skeletal muscle regeneration is attenuated in global Fn14-KO mice

We first investigated how the levels of Fn14 are regulated in the regenerating skeletal muscle of adult mice. One side tibialis anterior (TA) muscle of 8-wk-old WT mice was injured by intramuscular injection of 1.2% $BaCl_2$ solution, whereas the contralateral TA muscle was used as a control. To evaluate the initial phases of skeletal muscle regeneration, the TA muscle was isolated 5 d post-

injury and analyzed by performing quantitative real-time PCR (qRT–PCR) and Western blot. Consistent with published reports (Girgenrath et al, 2006; Mittal et al, 2010b), we found that mRNA and protein levels of Fn14 were significantly increased in the injured TA muscle compared with the contralateral uninjured muscle (Fig 1A and B). To understand the role of Fn14 in skeletal muscle regeneration, we first employed WT and whole-body Fn14-KO mice. The TA muscle of the mice was injured by intramuscular injection of 1.2% $BaCl_2$ solution, and muscle regeneration was evaluated 5 d later. There was no significant difference in wet weight of corresponding uninjured and injured TA muscles normalized by body weight between WT and Fn14-KO mice (Fig S1A). We next generated TA muscle sections and performed hematoxylin and eosin (H&E) staining. In agreement with a previously published report (Girgenrath et al, 2006), we found that muscle regeneration was considerably reduced in Fn14-KO mice compared with WT mice (Fig 1C). The average cross-sectional area (CSA) of centronucleated myofibers was significantly lower in the 5d-injured TA muscle of Fn14-KO mice compared with WT mice (Fig 1D and E). We also performed immunostaining for embryonic isoform of myosin heavy chain (eMyHC) protein that is expressed only in newly formed myofibers (Fig S1B). This analysis showed that the percentage of eMyHC$^+$ myofibers containing two or more centrally located nuclei was significantly reduced in the 5d-injured TA muscle of Fn14-KO mice compared with WT mice (Fig 1F). However, there was no difference in the total number of eMyHC$^+$ myofibers (Fig S1C). We also measured protein levels of various markers of muscle regeneration, such as MyoD, myogenin, and eMyHC. Results showed that there was no significant difference in the protein levels of MyoD between the two groups. However, protein levels of myogenin were significantly reduced in the injured TA muscle of Fn14-KO mice compared with WT. Intriguingly, we found a significant increase in the levels of eMyHC protein in the injured TA muscle of Fn14-KO mice compared with WT mice (Fig S1D and E). Because satellite cells play an indispensable role in muscle regeneration (Yin et al, 2013), we next investigated whether genetic ablation of Fn14 affects abundance of satellite cells in the skeletal muscle. Transverse sections generated from uninjured and 5d-injured TA muscles of WT and Fn14-KO mice were immunostained for Pax7 (a marker for satellite cells) and laminin (to mark myofiber boundaries) proteins, whereas nuclei were counterstained with DAPI (Fig 1G). There was no significant difference in the number of Pax7$^+$ cells in uninjured or injured TA muscles of WT and Fn14-KO mice (Fig 1H). Moreover, protein levels of Pax7 were comparable in the 5d-injured TA muscle of WT and Fn14-KO mice (Fig S1C and D). Western blot confirmed that Fn14 protein was absent in the 5d-injured TA muscle of Fn14-KO mice (Fig 1I). These results suggest that constitutive deletion of Fn14 attenuates skeletal muscle regeneration in adult mice.

## Myoblast-specific deletion of Fn14 inhibits muscle regeneration

A previously published report suggests that attenuation in muscle regeneration in whole-body Fn14-KO mice could be attributed to the diminished inflammatory response in the skeletal muscle in response to injury (Girgenrath et al, 2006). However, the cell-autonomous role of Fn14 in skeletal muscle regeneration remained unknown. We investigated whether targeted ablation of Fn14 in

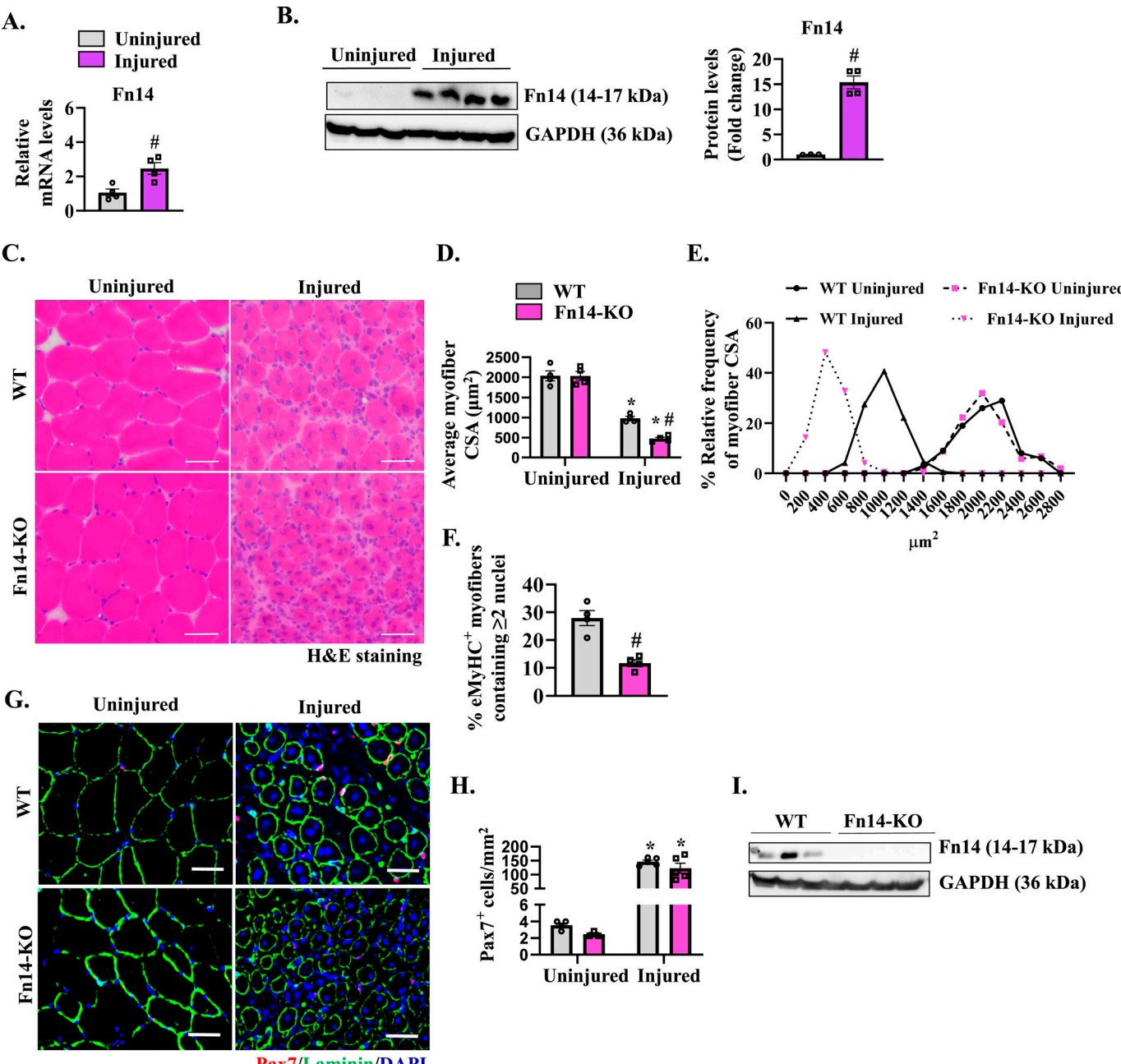

**Figure 1. Global ablation of Fn14 attenuates muscle regeneration in mice.**
WT mice were given an intramuscular injection of saline alone or 1.2% BaCl₂ solution in the TA muscle. After 5 d, the muscle was harvested and processed by qRT–PCR and Western blotting. **(A)** Relative mRNA levels of Fn14. n = 4. **(B)** Representative immunoblots and densitometry analysis showing the level of Fn14 protein in uninjured and injured TA muscles from WT mice. GAPDH was used as a loading control. n = 3. **(C)** Representative photomicrographs of uninjured and injured TA muscle transverse sections of WT and Fn14-KO mice after hematoxylin and eosin (H&E) staining. Scale bar: 50 μm. **(D)** Quantification of the average myofiber cross-sectional area with centralized nuclei. n = 4. **(E)** Relative frequency distribution of the myofiber cross-sectional area in uninjured and injured TA muscles of WT and Fn14-KO mice. **(F)** Percentage of eMyHC+ myofibers with two or more centrally located nuclei in 5d-injured TA muscle sections of WT and Fn14-KO mice. n = 4. **(G)** Representative photomicrographs of uninjured and 5d-injured TA muscle sections of WT and Fn14-KO mice after immunostaining for Pax7 (red) and laminin (green) proteins. Nuclei were identified by staining with DAPI. Scale bar: 50 μm. **(H)** Average number of Pax7+ cells per millimeter². n = 3–5. **(I)** Representative Western blots showing Fn14 levels in the 5d-injured TA muscle of WT and Fn14-KO mice. GAPDH was used as a loading control. Data are presented as the mean ± SEM analyzed by an unpaired t test or by a two-way ANOVA followed by Tukey's multiple-comparison test. *P ≤ 0.05, values significantly different from the corresponding uninjured muscle. #P ≤ 0.05, values significantly different from the corresponding muscle of WT mice.
Source data are available for this figure.

differentiated myofibers or myoblasts affects skeletal muscle regeneration in adult mice. To generate myofiber-specific Fn14-KO mice, we employed muscle creatine kinase (MCK)-Cre mice. Floxed Fn14 (henceforth Fn14$^{fl/fl}$) mice were crossed with MCK-Cre mice to generate Fn14$^{fl/fl}$; MCK-Cre (henceforth Fn14$^{mKO}$) and littermate Fn14$^{fl/fl}$ mice, as described previously (Tomaz da Silva et al, 2022). The TA muscle of 8-wk-old Fn14$^{fl/fl}$ and Fn14$^{mKO}$ mice was injured by intramuscular injection of 1.2% BaCl$_2$ solution, and muscle regeneration was studied at day 5 post-injury. There was no significant difference in overall body weight of Fn14$^{fl/fl}$ and Fn14$^{mKO}$ mice (Fig S2A). Moreover, there was no difference in wet weight of the corresponding uninjured or 5d-injured TA muscle normalized by body weight between Fn14$^{fl/fl}$ and Fn14$^{mKO}$ mice (Fig S2B). We next generated TA muscle transverse sections followed by performing H&E staining and morphometric analysis (Fig S2C). There was no significant difference in the average myofiber CSA or the percentage of myofibers containing two or more centrally located nuclei in the 5d-injured TA muscle of the two groups (Fig S2D–F). Furthermore, anti-eMyHC staining showed no difference in the expression of eMyHC in the regenerating TA muscle of Fn14$^{fl/fl}$ and Fn14$^{mKO}$ mice (Fig S2G), suggesting that myofiber-specific deletion of Fn14 does not affect skeletal muscle regeneration in adult mice.

We next sought to investigate whether myoblast-specific ablation of Fn14 affects skeletal muscle regeneration in adult mice. Fn14$^{fl/fl}$ mice were crossed with Myod1-Cre mice to generate myoblast-specific Fn14-KO (Fn14$^{fl/fl}$;Myod1-Cre; henceforth Fn14$^{myoKO}$) and littermate Fn14$^{fl/fl}$ mice. Next, the TA muscle of 8-wk-old littermate Fn14$^{fl/fl}$ and Fn14$^{myoKO}$ mice was injured by intramuscular injection of 1.2% BaCl$_2$ solution. Muscle regeneration was evaluated at day 5 or 14 post-injury. There was no significant difference in overall body weight or uninjured TA muscle weight normalized by body weight of Fn14$^{fl/fl}$ and Fn14$^{myoKO}$ mice (Fig 2A and B). However, on day 5 post-injury, there was a significant decrease in TA muscle wet weight normalized by body weight in both groups. Interestingly, wet weight of the 5d-injured TA muscle was significantly lower in Fn14$^{myoKO}$ mice compared with Fn14$^{fl/fl}$ mice (Fig 2B). In addition, there was a significant increase in TA muscle weight normalized by body weight, on day 14 post-injury, only in Fn14$^{fl/fl}$ mice compared with the uninjured muscle (Fig 2C). We next generated TA muscle transverse sections and performed H&E staining (Fig 2D). Intriguingly, the size of regenerating myofibers was significantly reduced and there was an increase in mononuclear cell infiltration between myofibers in the TA muscle of Fn14$^{myoKO}$ mice compared with Fn14$^{fl/fl}$ mice at both day 5 and day 14 post-injury (Fig 2D–G). In addition, there was a significant reduction in the percentage of myofibers containing two or more centrally located nuclei in the 5d-injured TA muscle of Fn14$^{myoKO}$ mice compared with corresponding Fn14$^{fl/fl}$ mice (Fig 2H). Myod1-Cre is a knock-in line in which Cre recombinase cDNA has been inserted at one of the alleles of the Myod1 gene (Chen et al, 2005). Because Myod1 is a critical regulator of myogenesis, we investigated whether the lack of one allele of Myod1 contributes to the observed phenotype in myoblast-specific Fn14-KO mice. To address this issue, we compared muscle regeneration defects in Fn14$^{myoKO}$ mice with the mice that have the Myod1-Cre allele but heterozygous for the floxed Fn14 allele (i.e., Fn14$^{fl/wt}$; Myod1-Cre). Results showed that there was a significant deficit in muscle regeneration in Fn14$^{myoKO}$ mice

compared with corresponding Fn14$^{fl/wt}$; Myod1-Cre mice at day 5 post-injury, confirming that myoblast Fn14 is essential for muscle regeneration (Fig S3A and B).

Two consecutive injuries carried out 3 or 4 wk apart, with enough time elapse allowing the regeneration of the muscle after the first injury, is another model to study muscle progenitor cell function, maintenance, or depletion (Hardy et al, 2016; Roy et al, 2021). Therefore, we next examined muscle regeneration in Fn14$^{fl/fl}$ and Fn14$^{myoKO}$ mice after performing double injury. On day 21 after the first injury, the TA muscle was injured again by intramuscular injection of 1.2% BaCl$_2$ solution. At day 5 after the second injury, the muscle was isolated and analyzed by performing H&E staining (Fig 2I). Our analysis showed that the average myofiber CSA and the percentage of myofibers containing two or more centrally located nuclei were significantly reduced in Fn14$^{myoKO}$ mice compared with Fn14$^{fl/fl}$ mice upon double injury (Fig 2J and K). Taken together, these results suggest that Fn14-mediated signaling in myoblasts but not in myofibers is essential for efficient muscle regeneration.

### Myoblast-specific ablation of Fn14 reduces myofiber formation

To understand the mechanisms by which myoblast Fn14 regulates skeletal muscle regeneration, we studied the expression of early markers of muscle regeneration. We first performed immunostaining for eMyHC and laminin proteins on 5d-injured TA muscle sections. Nuclei were counterstained with DAPI (Fig 3A). Results showed that the percentage of eMyHC-positive myofibers containing two or more centrally located nuclei was significantly reduced in Fn14$^{myoKO}$ mice compared with Fn14$^{fl/fl}$ mice (Fig 3B). In a parallel experiment, uninjured and 5d-injured TA muscles of Fn14$^{fl/fl}$ and Fn14$^{myoKO}$ mice were analyzed by performing qRT–PCR and Western blot. There was a significant increase in the mRNA and protein levels of eMyHC, MyoD, and myogenin in the injured TA muscle compared with the contralateral uninjured muscle in both groups (Fig 3C–E). However, mRNA levels of eMyHC, MyoD, and myogenin, as well as protein levels of myogenin, were significantly higher in injured TA muscle of Fn14$^{myoKO}$ mice compared with Fn14$^{fl/fl}$ mice (Fig 3C–E). To determine whether myoblast-specific deletion of Fn14 affects the number of satellite cells, transverse sections generated from the 5d-injured TA muscle of Fn14$^{fl/fl}$ and Fn14$^{myoKO}$ mice were immunostained for Pax7 and laminin proteins. Nuclei were identified by staining with DAPI (Fig 3F). Interestingly, the number of Pax7$^+$ cells per unit area was significantly higher in the injured TA muscle of Fn14$^{myoKO}$ mice compared with Fn14$^{fl/fl}$ mice (Fig 3G). In addition, mRNA levels of Pax7 were significantly increased in the 5d-injured TA muscle of Fn14$^{myoKO}$ mice compared with Fn14$^{fl/fl}$ mice (Fig 3H). These results suggest that although deletion of Fn14 in myoblasts reduces myofiber formation, it does not impede the expression of various MRFs and abundance of satellite cells in the regenerating skeletal muscle of adult mice.

### Fn14 promotes myoblast fusion in regenerating myofibers

Because ablation of Fn14 in myoblasts does not reduce the number of satellite cells or the expression of various MRFs in the regenerating muscle, we next investigated whether Fn14 has any role in regulating myoblast fusion during muscle regeneration. For

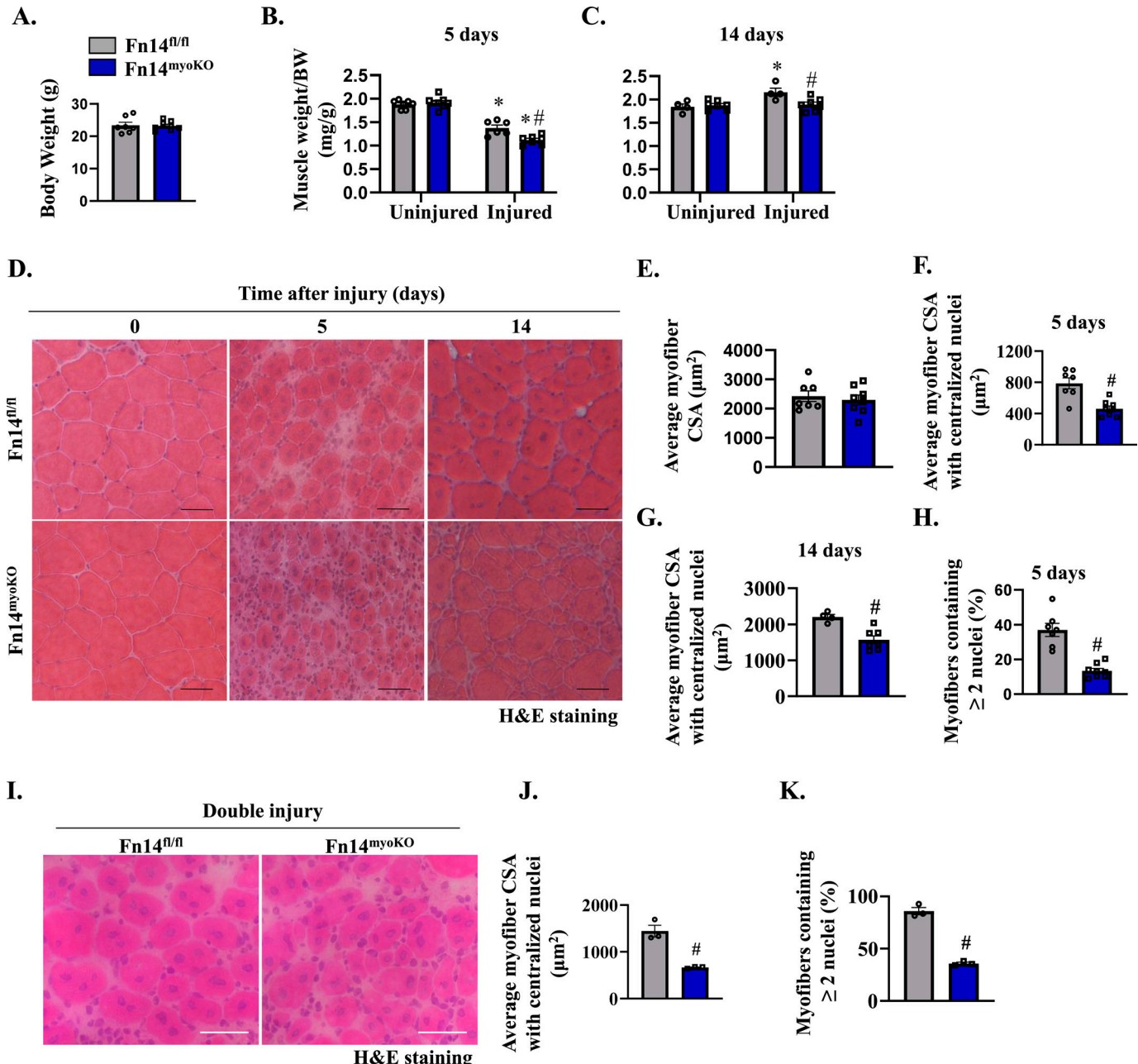

**Figure 2. Myoblast-specific ablation of Fn14 inhibits muscle regeneration in mice.**
**(A)** Average body weight of 8-wk-old littermate Fn14$^{fl/fl}$ and Fn14$^{myoKO}$ mice. **(B, C)** Wet weight of uninjured and injured TA muscles at (B) day 5 and (C) day 14 normalized by body weight of Fn14$^{fl/fl}$ and Fn14$^{myoKO}$ mice. n = 6–8 per group. **(D)** Representative photomicrographs of hematoxylin-and-eosin (H&E)–stained sections at days 0, 5, and 14 after intramuscular injection of BaCl$_2$ in the TA muscle of Fn14$^{fl/fl}$ and Fn14$^{myoKO}$ mice. Scale bar: 50 $\mu$m. **(E)** Quantification of the average myofiber cross-sectional area (CSA) in the uninjured TA muscle of Fn14$^{fl/fl}$ and Fn14$^{myoKO}$ mice. **(F, G)** Average myofiber CSA with centralized nuclei (F) at day 5 post-injury (n = 7–8) and (G) at day 14 post-injury (n = 4–7). **(H)** Percentage of myofibers containing two or more centrally located nuclei in the 5d-injured TA muscle of Fn14$^{fl/fl}$ and Fn14$^{myoKO}$ mice. n = 7–8 per group. After 21 d of the first injury, the TA muscle of Fn14$^{fl/fl}$ and Fn14$^{myoKO}$ mice was again given an intramuscular injection of 50 $\mu$l of 1.2% BaCl$_2$ solution, and the muscle was analyzed at day 5. **(I, J, K)** Representative photomicrograph of H&E-stained TA muscle sections, and quantification of (J) the average myofiber CSA and (K) the percentage of myofibers containing two or more centrally located nuclei. Scale bar: 50 $\mu$m. n = 3 per group. Data are presented as the mean ± SEM analyzed by an unpaired $t$ test or by a two-way ANOVA followed by Tukey's multiple-comparison test. *$P \leq 0.05$, values significantly different from the corresponding uninjured muscle. #$P \leq 0.05$, values significantly different from the corresponding muscle of Fn14$^{fl/fl}$ mice.

this experiment, the TA muscle of 8-wk-old Fn14$^{fl/fl}$ and Fn14$^{myoKO}$ mice was injured by intramuscular injection of 1.2% BaCl$_2$ solution. After 72 h, the mice were given a single intraperitoneal injection of

EdU and the TA muscle was isolated 11 d later. Finally, transverse muscle sections were generated and processed for the detection of EdU$^+$ myonuclei (Fig 4A). Results showed that the number of EdU$^+$

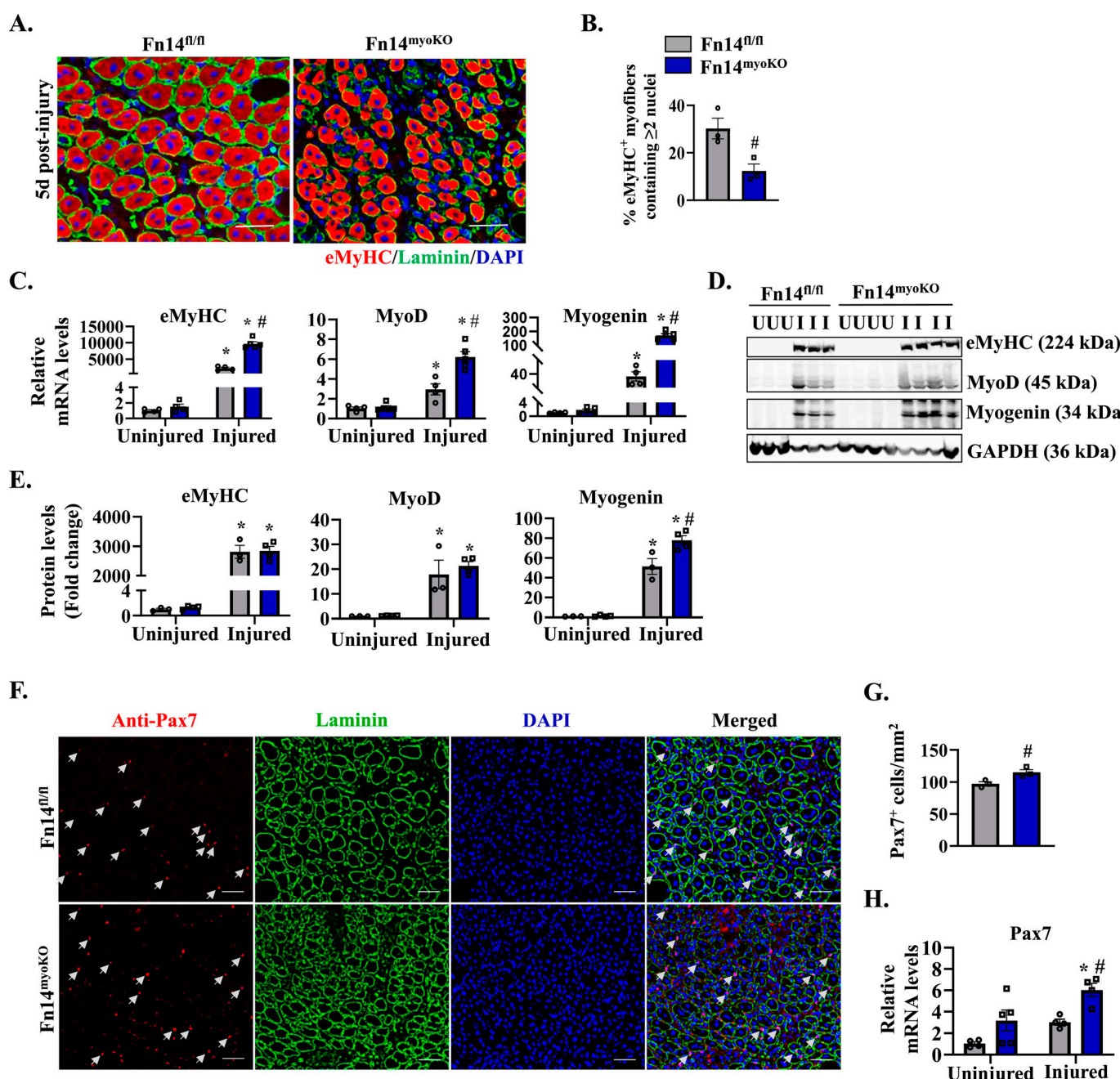

**Figure 3. Deletion of Fn14 in myoblasts reduces myofiber regeneration in adult mice.**
**(A)** Representative photomicrographs of 5d-injured TA muscle sections of Fn14^fl/fl and Fn14^myoKO mice after immunostaining for eMyHC (red) and laminin (green). Nuclei were counterstained with DAPI (blue). Scale bar: 50 μm. **(B)** Percentage of eMyHC⁺ myofibers with two or more centrally located nuclei in 5d-injured TA muscle sections of Fn14^fl/fl and Fn14^myoKO mice. n = 3 per group. Data are presented as the mean ± SEM. #P ≤ 0.05, values significantly different from the corresponding injured TA muscle of Fn14^fl/fl mice by an unpaired t test. **(C)** Relative mRNA levels of eMyHC, MyoD, and myogenin in uninjured and 5d-injured TA muscles of Fn14^fl/fl and Fn14^myoKO mice assayed by performing qRT–PCR. n = 4–5 per group. **(D, E)** Immunoblots and (E) densitometry analysis showing levels of eMyHC, MyoD, and myogenin proteins in uninjured and 5d-injured TA muscles from Fn14^fl/fl and Fn14^myoKO mice. GAPDH was used as a loading control. n = 3–4 per group. **(F)** Representative photomicrographs of uninjured and 5d-injured TA muscle sections of Fn14^fl/fl and Fn14^myoKO mice after immunostaining for Pax7 (red) and laminin (green). Nuclei were identified by staining with DAPI. Scale bar: 50 μm. **(G)** Average number of Pax7-positive cells per millimeter². n = 3 per group. **(H)** Relative mRNA levels of Pax7 of uninjured and 5d-injured TA muscles of Fn14^fl/fl and Fn14^myoKO mice. n = 4–5 per group. Data are presented as the mean ± SEM analyzed by an unpaired t test or by a two-way ANOVA followed by Tukey's multiple-comparison test. *P ≤ 0.05, values significantly different from the corresponding uninjured muscle. #P ≤ 0.05, values significantly different from the corresponding muscle of Fn14^fl/fl mice.

Source data are available for this figure.

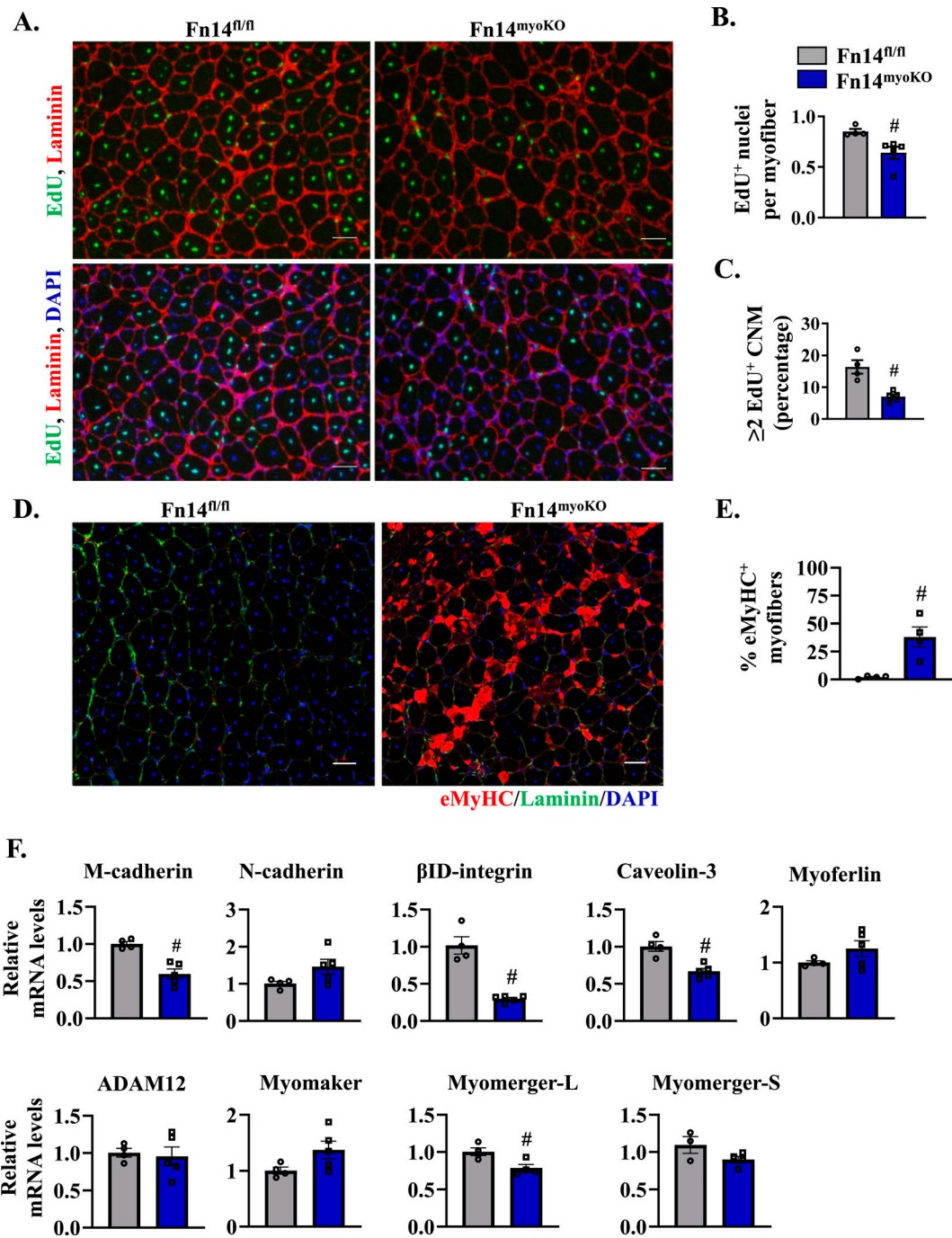

**Figure 4. Fn14 mediates myoblast fusion during skeletal muscle regeneration.**
The TA muscle of 8-wk-old Fn14$^{fl/fl}$ and Fn14$^{myoKO}$ mice was injured by intramuscular injection of 50 $\mu$l of 1.2% BaCl$_2$ solution. After 3 d, the mice were given an intraperitoneal injection of EdU. After 11 d, the TA muscle was collected and muscle sections prepared were stained to detect EdU, laminin, and nuclei. **(A)** Representative photomicrographs after EdU (green), laminin (red), and DAPI (blue) staining are presented here. Scale bar: 50 $\mu$m. **(B)** Quantification of the percentage of EdU$^+$ nuclei per myofiber. n = 4–5 per group. **(C)** Percentage of myofibers containing two or more EDU$^+$ centrally located nuclei in the TA muscle of Fn14$^{fl/fl}$ and Fn14$^{myoKO}$ mice. n = 4–5 per group. **(D)** Representative photomicrographs of 14d-injured TA muscle sections of Fn14$^{fl/fl}$ and Fn14$^{myoKO}$ mice after immunostaining for eMyHC (red) and laminin (green). Nuclei were counterstained with DAPI (blue). Scale bar: 50 $\mu$m. **(E)** Percentage of eMyHC$^+$ myofibers with two or more centrally located nuclei in 14d-injured TA muscle sections of Fn14$^{fl/fl}$ and Fn14$^{myoKO}$ mice. n = 4 per group. **(F)** Relative mRNA levels of various profusion molecules assayed by performing qRT–PCR in the 5d-injured TA muscle of Fn14$^{fl/fl}$ and Fn14$^{myoKO}$ mice. n = 3–5 per group. Data are presented as the mean ± SEM analyzed by an unpaired t test. #$P \leq 0.05$, values significantly different from the corresponding injured TA muscle of Fn14$^{fl/fl}$ mice.

nuclei per myofiber was significantly reduced in the TA muscle of Fn14[myoKO] mice compared with Fn14[fl/fl] mice (Fig 4B). In addition, the percentage of myofibers containing two or more EdU⁺ nuclei was also significantly reduced in the TA muscle of Fn14[myoKO] mice compared with Fn14[fl/fl] mice (Fig 4C). Impairment in myoblast fusion results in the persistent expression of eMyHC in regenerating myofibers. We next performed immunostaining for eMyHC protein in the TA muscle at day 14 after injury (Fig 4D). There were almost no eMyHC⁺ myofibers in the 14d-injured TA muscle of Fn14[fl/fl] mice suggesting normal progression of muscle regeneration. In contrast, small-sized eMyHC⁺ myofibers were in abundance in the 14d-injured TA muscle of Fn14[myoKO] mice (Fig 4D and E).

Myoblast fusion exhibits its peak between 4 and 7 d after muscle injury followed by maturation and functional recovery phases (Forcina et al, 2020). To understand the molecular mechanisms by which Fn14 regulates myoblast fusion in vivo, we also measured transcript levels of a few profusion molecules in the 5d-injured TA muscle of Fn14[fl/fl] and Fn14[myoKO] mice. Results showed that mRNA levels of M-cadherin, caveolin-3, $\beta$1D-integrin, and myomerger-L were significantly reduced in the 5d-injured TA muscle of Fn14[myoKO] mice compared with Fn14[fl/fl] mice. In contrast, there was no significant difference in mRNA levels of N-cadherin, myoferlin, myomaker, ADAM12, and myomerger-S in the 5d-injured TA muscle of Fn14[fl/fl] and Fn14[myoKO] mice (Fig 4F). These results suggest that Fn14 mediates myoblast fusion during muscle regeneration in adult mice.

### Fn14 coordinates the differentiation and fusion of cultured myoblasts

To investigate the role and mechanisms of action of Fn14 during myogenesis, we established primary myoblast cultures from WT and whole-body Fn14-KO mice. We used myoblasts isolated from whole-body Fn14-KO mice rather than Fn14[myoKO] mice to mitigate any potential effect of the loss of the expression of Myod1 from one allele on myogenesis. Using RNA-Seq, we were able to define the effect of Fn14 deficiency on global gene expression in cultured myoblasts. The loss of Fn14 resulted in the differential expression of 1,383 genes by over 1.5-fold (computed using a FDR of $P < 0.05$) in Fn14-KO versus WT myoblasts. Of the 1,383 genes, 499 genes were down-regulated, whereas 884 genes were up-regulated in Fn14-KO compared with WT myoblasts (Fig 5A). Functional enrichment analysis of the differentially expressed genes using gene ontology (GO) annotations showed that several gene sets related to morphogenesis, migration, motility, and movement of subcellular components were down-regulated in Fn14-KO compared with WT cultures. In contrast, the gene sets involved in muscle cell development, differentiation, and regulation of immune response were up-regulated in Fn14-KO cultures (Fig 5B). Similar results were obtained when up-regulated and down-regulated gene sets were evaluated using the Metascape gene annotation and analysis tool (Fig S4). Interestingly, RNA-Seq analysis showed the higher expression of myogenin in Fn14-KO cultures compared with WT cultures in a normal growth medium (Fig 5C). We next investigated how the loss of Fn14 affects myogenic differentiation and myotube formation in vitro. WT and Fn14-KO myoblasts were incubated in DM

for 6, 24, or 48 h, and the cultures were immunostained for myosin heavy chain (MyHC), a marker of muscle differentiation (Fig 5D). Results showed that the number of MyHC⁺-mononucleated cells was significantly higher in Fn14-KO cultures compared with WT cultures at all the time points after addition of DM (Fig 5E). Importantly, the proportion of myotubes containing 2–4 nuclei was significantly increased, whereas those containing 5–10 or more than 10 nuclei were significantly reduced in Fn14-KO cultures compared with corresponding WT cultures at 24 or 48 h of incubation in differentiation medium (Fig 5F). To understand whether Fn14 affects myogenic differentiation, we measured protein levels of myogenin and MyHC at different time points after addition of DM. Consistent with RNA-Seq results, there was a slight increase in the levels of myogenin and MyHC in Fn14-KO cultures compared with WT cultures at 0 and 6 h after addition of DM. However, protein levels of both myogenin and MyHC were comparable in WT and Fn14-KO cultures at 24, 48, and 72 h after addition of DM (Figs 5G and S5). These results suggest that the loss of Fn14 leads to precocious differentiation of myoblasts, but myotube formation is diminished because of the inhibition of myoblast fusion.

There are published reports suggesting that the Fn14 receptor can function independent of its ligand TWEAK (Winkles, 2008; Burkly, 2014; Tajrishi et al, 2014). Because deletion of Fn14 inhibits myotube formation, we next sought to determine whether the forced expression of Fn14 in myoblasts is sufficient to improve myotube formation after induction of differentiation. Primary myoblasts were transduced with the retrovirus expressing EGFP or Fn14 protein. The cells were then incubated in DM for 48 h, and the myotube formation was examined by performing anti-MyHC and DAPI staining (Fig 5H). Interestingly, the overexpression of Fn14 improved myotube formation. The average myotube diameter was significantly higher in Fn14-overexpressing cultures compared with control cultures (Fig 5I). We also investigated whether the forced expression of Fn14 affects the expression of various MRFs and MyHC. Although there was no difference in MyoD, the levels of myogenin were significantly higher in Fn14-overexpressing cultures compared with controls in the growth medium. However, the levels of MyoD or myogenin were comparable between control and Fn14-overexpressing cultures after 48 h of incubation in DM. Intriguingly, we found that the levels of MyHC were significantly higher in Fn14-overexpressing cultures compared with controls at 48 h of addition of DM, suggesting that the overexpression of Fn14 augments myogenic differentiation (Fig S6A and B). The p38 MAPK signaling and Akt signaling positively regulate initial stages of myogenic differentiation (Perdiguero et al, 2007; Gardner et al, 2012; Brennan et al, 2021). Therefore, we investigated whether the forced expression of Fn14 affects activation of p38 MAPK or Akt in cultured myoblasts. Results showed that the levels of phosphorylated p38 (p-p38) and phosphorylated Akt (p-Akt) were significantly higher in Fn14-overexpressing cultures compared with control cultures incubated in the normal growth medium. However, the levels of p-p38 or p-Akt were comparable between control and Fn14-overexpressing (Fn14-OE) myoblast cultures at 48 h of incubation in DM (Fig S6C–E). Western blot analysis confirmed increased levels of Fn14 in cultures transduced with the Fn14 retrovirus (Fig S6A and B). Altogether, these results suggest

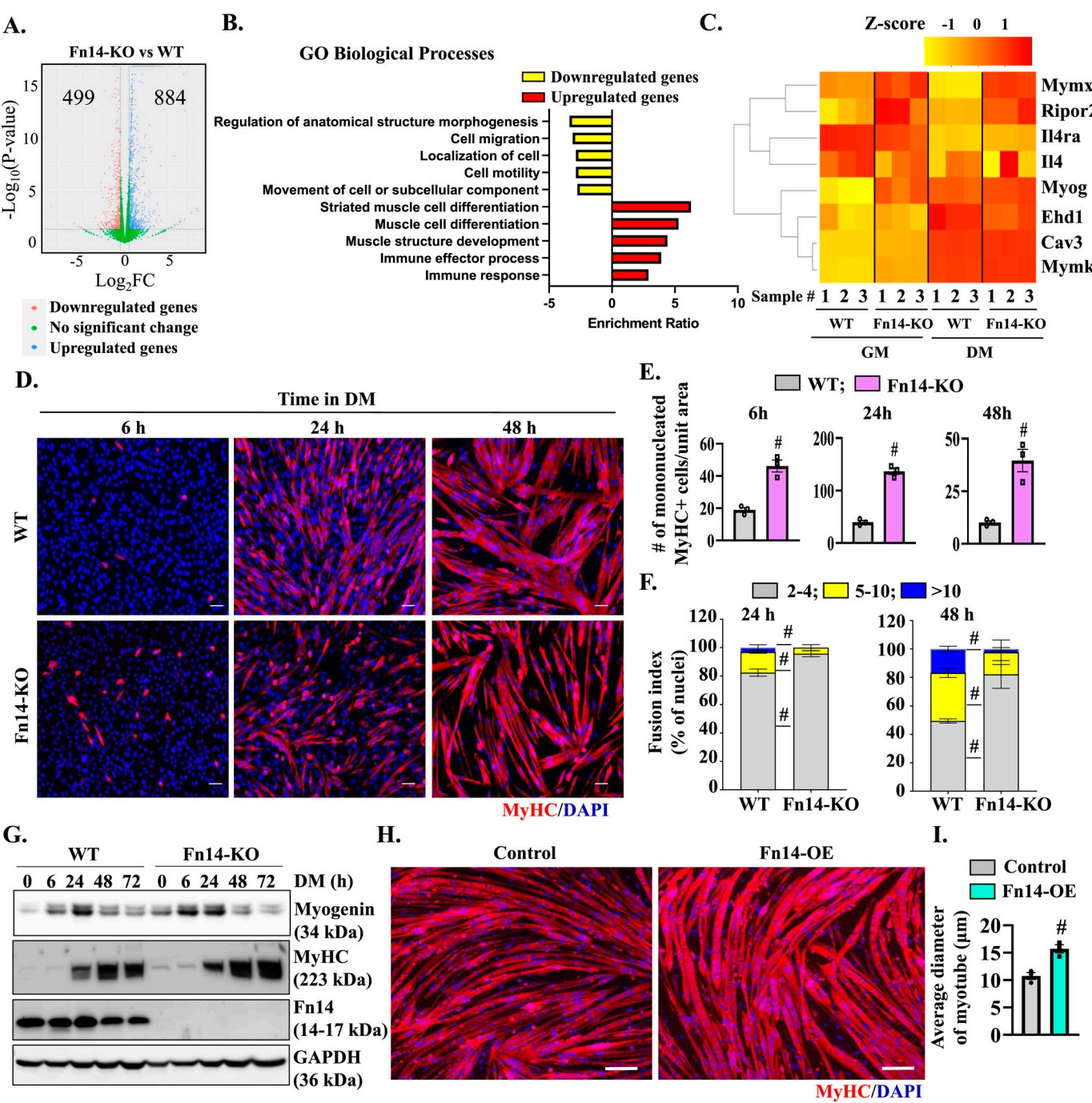

**Figure 5. Fn14 mediates myoblast fusion in vitro.**

Primary myoblasts isolated from hindlimb muscles of WT and Fn14-KO mice were plated at equal densities and incubated in the growth medium or the differentiation medium for 48 h followed by RNA-Seq analysis. **(A)** Volcano plot from RNA-Seq analysis of myoblasts of Fn14-KO mice versus WT illustrating down-regulated (red dots) and up-regulated (blue dots) genes with a threshold of $log_2FC \geq |0.5|$ and $P$-value $\leq 0.05$. **(B)** Gene ontology (GO) biological processes associated with down-regulated and up-regulated genes. **(C)** Heatmap showing regulation of selected genes involved in myogenesis and myoblast fusion. **(D)** Primary myoblasts of WT and Fn14-KO mice were plated at equal densities and incubated in DM for 6, 24, or 48 h followed by staining for MyHC (red) and DAPI (blue). Photomicrographs are presented here. Scale bar: 50 $\mu$m. **(E)** Quantification of the number of mononucleated MyHC$^+$ cells per unit area (~1.75 mm$^2$) in WT and Fn14-KO cultures after 6, 24, or 48 h of incubation in DM. n = 3 per group. **(F)** Quantification of the fusion index in WT and Fn14-KO cultures after 24 or 48 h of addition of DM. We measured the percentage of MyHC$^+$ nuclei that were inside the myotubes containing between 2 and 4 nuclei (2–4), the percentage of MyHC$^+$ nuclei that were inside the myotubes containing between 5 and 10 nuclei (5–10), and the percentage of MyHC$^+$ nuclei that were inside the myotubes containing more than 10 nuclei (>10). n = 3 in each group. **(G)** Immunoblots showing protein levels of myogenin, MyHC, Fn14, and GAPDH in WT and Fn14-KO cultures at 0, 6, 24, 48, and 72 h after incubation in DM. **(H)** Primary myoblasts prepared from WT mice were transduced with the retrovirus expressing EGFP (control) or Fn14 and incubated in DM for 48 h followed by staining for MyHC protein. Nuclei were stained with DAPI. Scale bar: 50 $\mu$m.

that Fn14 promotes myotube formation through stimulating myoblast fusion and fine-tuning myogenic differentiation.

### Fn14 regulates the expression of components of calcium-dependent signaling

To understand the signaling mechanisms by which Fn14 promotes myotube formation, we measured the phosphorylation of various signaling molecules that are known to promote myoblast fusion. WT and Fn14-KO myoblasts were incubated in DM for different time periods, and the cell extracts prepared were analyzed by performing Western blot. Results showed that there was no difference in the levels of phosphorylated ERK1/2 and ERK5 between WT and Fn14-KO cultures. Furthermore, the levels of phosphorylated p65 (a marker of canonical NF-κB signaling) and relative levels of p100/p52 (a marker of non-canonical NF-κB signaling) were comparable in WT and Fn14-KO cultures at different time points after addition of DM (Fig 6A). Similarly, there was no significant difference in the levels of phosphorylated ERK1/2 and phosphorylated p65, or in relative amounts of p100/p52 protein in the 5d-injured TA muscle of Fn14$^{fl/fl}$ and Fn14$^{myoKO}$ mice (Fig 6B and C). Published reports suggest that the calcium (Ca$^{2+}$)-dependent signaling pathway also plays an important role in myoblast fusion during myogenesis (Constantin et al, 1996; Bijlenga et al, 2000; Hindi et al, 2013). Interestingly, the levels of NFATc2 protein, a component of Ca$^{2+}$–calcineurin–NFAT pathway, were considerably reduced in Fn14-KO cultures compared with WT cultures (Fig 6A). Moreover, transcript levels of various regulators of calcium-dependent signaling, such as calpain-3, calsequestrin-1, calsequestrin-2, RyR1, and CamKIIb, but not calmodulin1, were significantly reduced in Fn14-KO myoblast cultures compared with WT cultures (Fig 6D). We also studied the expression of these molecules in the regenerating muscle of Fn14$^{fl/fl}$ and Fn14$^{myoKO}$ mice. Results showed that there was a significant reduction in the mRNA levels of calpain-3, calsequestrin-1, calsequestrin-2, and CamKIIb in the 5d-injured TA muscle of Fn14$^{myoKO}$ mice compared with Fn14$^{fl/fl}$ mice (Fig 6E). Collectively, these results suggest that Fn14-mediated signaling regulates the expression of various molecules involved in the regulation of calcium-dependent pathways.

### Fn14 stimulates Wnt signaling in myoblasts

Our RNA-Seq analysis showed that the expression of a few components of Wnt signaling is differentially regulated between WT and Fn14-KO cultured myoblasts (Fig 7A). Interestingly, canonical Wnt signaling also plays a major role in myoblast fusion (Abmayr & Pavlath, 2012; Hindi et al, 2013). To understand the impact of Fn14 on Wnt signaling, we measured the levels of Wnt3a protein in WT and Fn14-KO cultures at different time points after addition of DM. Intriguingly, the protein levels of Wnt3a were increased in WT cultures but not in Fn14-KO cultures after addition of DM (Fig 7B). Phosphorylation of GSK-3β at Ser9 residue leads to its inactivation, which is required for activation of canonical Wnt signaling (Tejeda-Munoz & Robles-Flores, 2015). We found that the levels of phosphorylated GSK-3β (Ser9) were reduced in Fn14-KO cultures compared with WT cultures (Fig 7B). We previously reported that MyD88 protein regulates Wnt signaling to promote myoblast fusion during myogenesis (Hindi et al, 2017b). Accordingly, we found that the levels of MyD88 were also reduced in differentiating Fn14-KO cultures compared with WT cultures (Fig 7B). By performing qRT–PCR, we next investigated how the expression of various components of Wnt signaling is regulated in the regenerating skeletal muscle of Fn14$^{myoKO}$ mice. Interestingly, the mRNA levels of Wnt ligands Wnt4, Wnt5a, and Wnt11; Wnt receptor Frizzled-4 (Fzd4); and Wnt target gene Axin-2 were found to be significantly reduced in the 5d-injured TA muscle of Fn14$^{myoKO}$ mice compared with the corresponding injured TA muscle of Fn14$^{fl/fl}$ mice (Fig 7C–E).

We also studied the effects of the overexpression of Fn14 on the expression of components of Wnt signaling in myoblasts. Primary WT myoblasts were transduced with control or Fn14-expressing retrovirus followed by performing qRT–PCR. Interestingly, the mRNA levels of Wnt4, Wnt5a, Fzd1, Fzd2, Fzd4, and Axin-2 were significantly increased in Fn14-overexpressing myoblasts compared with corresponding controls (Fig 7F–H). These results suggest that Fn14-mediated signaling cross-talks with Wnt signaling during myogenesis.

### Forced activation of Wnt signaling rescues fusion defects in Fn14-KO cultures

Because the loss of Fn14 results in the inhibition of Wnt signaling, we next investigated whether forced activation of Wnt signaling improves fusion in Fn14-KO cultures. For this experiment, WT and Fn14-KO myoblasts were incubated in DM with or without recombinant Wnt3a protein for 48 h. In parallel, the cultures were also treated with LiCl, which activates canonical Wnt signaling through inhibiting GSK-3β (Lindsley et al, 2006). Finally, the cultures were fixed and the myotube formation was evaluated by immunostaining for MyHC protein (Fig 8A). Interestingly, addition of Wnt3a protein or LiCl significantly improved myotube formation in Fn14-KO cultures. Quantitative analysis confirmed that Wnt3a and LiCl significantly increased the proportion of myotubes containing 10 or more nuclei (Fig 8B). Furthermore, the average diameter of myotubes in Fn14-KO cultures was significantly higher in cultures treated with Wnt3a protein compared with corresponding controls (Fig 8C). These results suggest that Fn14 promotes myotube formation, at least in part, through activation of canonical Wnt signaling.

## Discussion

The Fn14 receptor is expressed on the cell surface of a variety of mesenchymal cells, including muscle progenitor cells (Girgenrath

(I) Average diameter of myotubes in control and Fn14-overexpressing (OE) cultures after 48 h of addition of DM. n = 3 per group. Data are presented as the mean ± SEM analyzed by an unpaired $t$ test. #$P ≤ 0.05$, values significantly different from the corresponding WT or control myotube by an unpaired $t$ test. Source data are available for this figure.

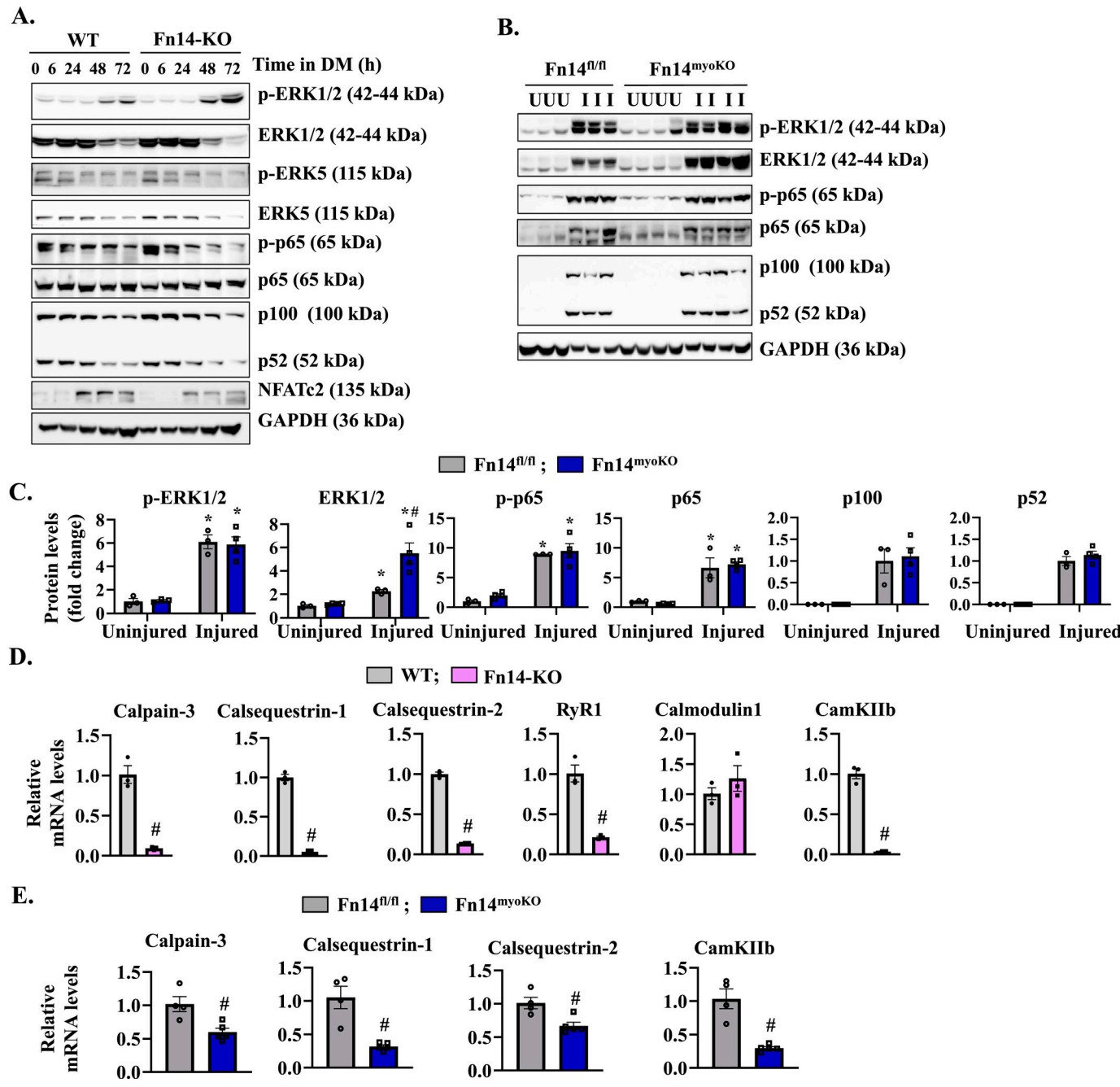

**Figure 6. Fn14 regulates components of calcium signaling.**
**(A)** Primary myoblasts prepared from WT and Fn14-KO mice were incubated in DM, and samples were collected at indicated time points. Representative immunoblots showing levels of phosphorylated and total ERK1/2, phosphorylated and total ERK5, phosphorylated and total p65, and total p100, p52, NFATc2, and unrelated protein GAPDH in WT and Fn14-KO cultures. **(B, C)** Representative immunoblots, and (C) densitometry analysis showing levels of phosphorylated and total ERK1/2, phosphorylated and total p65, and total p100 and p52 protein levels in uninjured and 5d-injured TA muscles of Fn14[fl/fl] and Fn14[myoKO] mice. GAPDH was used as a loading control. n = 3–4 per group. Data are presented as the mean ± SEM analyzed by a two-way ANOVA followed by Tukey's multiple-comparison test. **(D)** Relative mRNA levels of calpain-3, calsequestrin-1 and calsequestrin-2, RyR1, calmodulin1, and CamKIIb in WT and FN14-KO myoblast cultures. n = 3 per group. **(E)** Relative mRNA levels of calpain-3, calsequestrin-1 and calsequestrin-2, and CamKIIb in the 5d-injured TA muscle of Fn14[fl/fl] and Fn14[myoKO] mice. n = 3–5 per group. Data are presented as the mean ± SEM analyzed by an unpaired t test or by a two-way ANOVA followed by Tukey's multiple-comparison test. *P ≤ 0.05, values significantly different from the corresponding uninjured muscle. #P ≤ 0.05, values significantly different from the corresponding WT myoblast or Fn14[fl/fl] mice.
Source data are available for this figure.

et al, 2006; Dogra et al, 2007b; Burkly, 2014). Although the TWEAK cytokine engages Fn14 to induce different cellular responses, there are also reports suggesting that the Fn14 receptor can regulate

activation of downstream signaling pathways independent of the TWEAK cytokine (Winkles, 2008; Burkly, 2014; Tajrishi et al, 2014). The role of Fn14 in skeletal muscle regeneration was previously

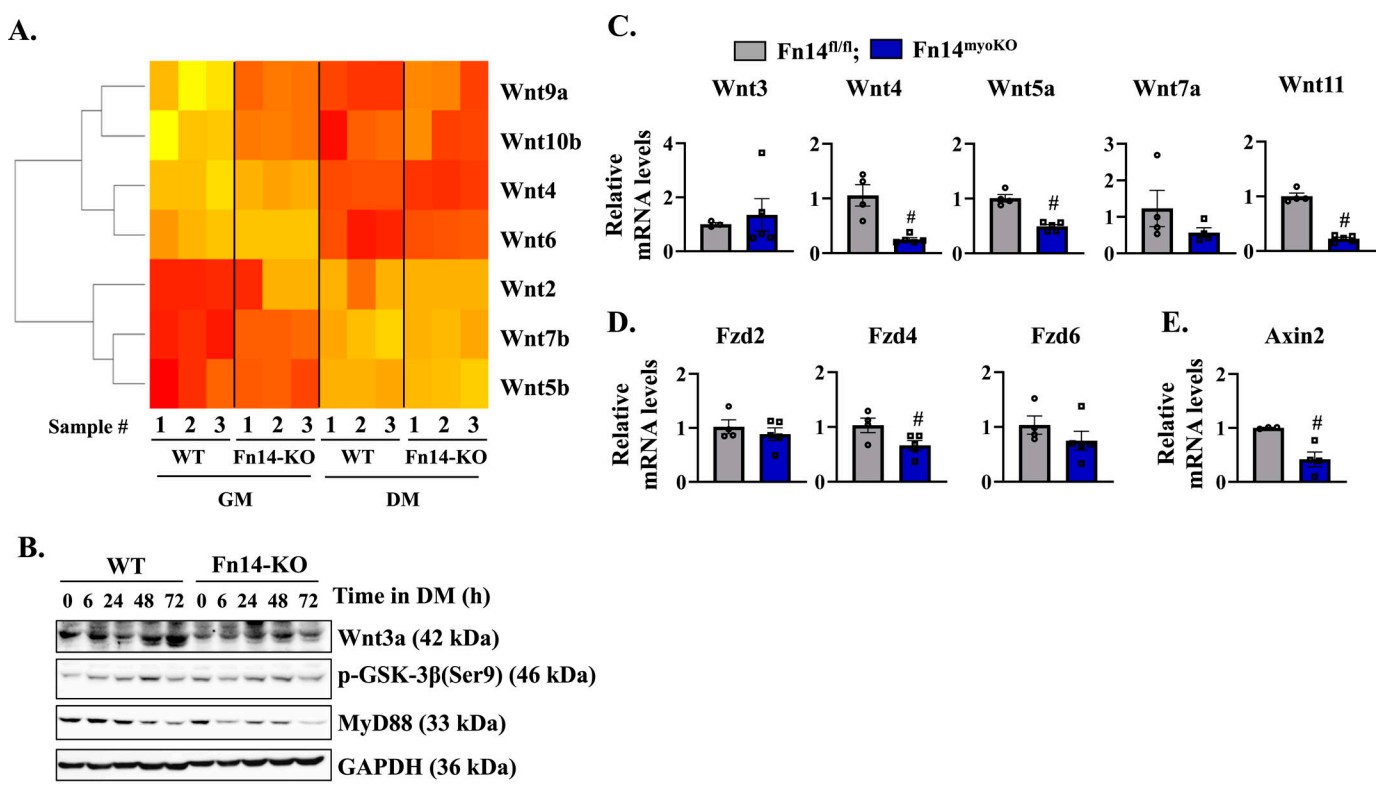

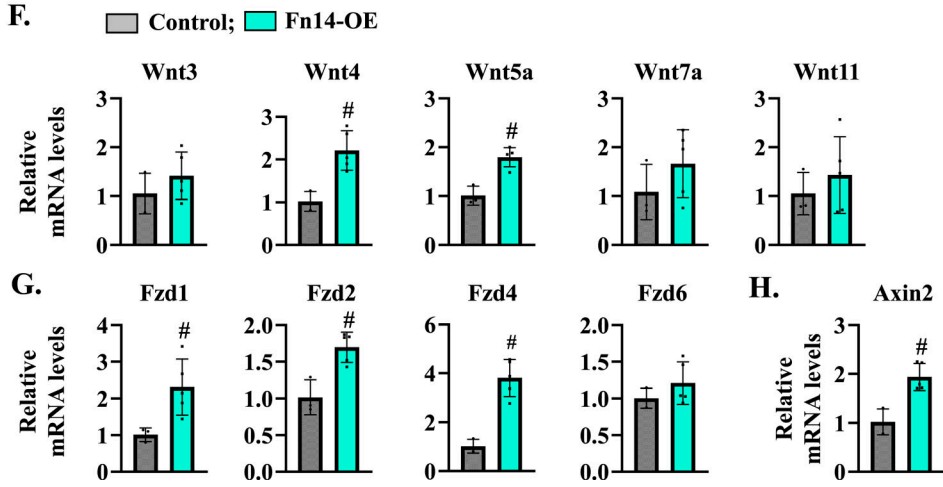

**Figure 7. Fn14 regulates Wnt signaling during myogenesis.**
**(A)** Heatmap showing the expression of genes involved in Wnt signaling in WT and Fn14-KO mice incubated in the growth medium (GM) or the differentiation medium (DM) for 48 h. **(B)** Representative immunoblots showing levels of Wnt3a, phospho-GSK-3$\beta$, MyD88, and unrelated protein GAPDH in WT and Fn14-KO cultures at different time points after addition of DM. **(C, D, E)** Relative mRNA levels of (C) Wnt ligands: Wnt3, Wnt4, Wnt5A, Wnt7A, and Wnt11; (D) Wnt receptors: Fzd2, Fzd4, and Fzd6; and (E) Wnt target gene: Axin-2 in the 5d-injured TA muscle of Fn14^{fl/fl} and Fn14^{myoKO} mice. n = 3–5 per group. **(F, G, H)** Relative mRNA levels of (F) Wnt ligands: Wnt3, Wnt4, Wnt5A, Wnt7A, and Wnt11; (G) Wnt receptors: Fzd1, Fzd2, Fzd4, and Fzd6; and (H) Wnt target: Axin-2 in control and Fn14-OE cultured myoblasts. n = 3 per group. Data are presented as the mean ± SEM analyzed by an unpaired $t$ test. #$P \le 0.05$, values significantly different from the corresponding injured TA muscle of Fn14^{fl/fl} mice or control myoblast cultures.
Source data are available for this figure.

evaluated using whole-body KO mice (Girgenrath et al, 2006). It was reported that muscle regeneration was delayed in Fn14-KO mice potentially because of inhibition in the recruitment of phagocytic cells after muscle damage (Girgenrath et al, 2006). However, the

cell-autonomous role of Fn14 in skeletal muscle regeneration remained unknown. In this study, using global and conditional Fn14-KO mice we have discovered that Fn14-mediated signaling regulates the myoblast fusion step during regenerative myogenesis.

**A.**

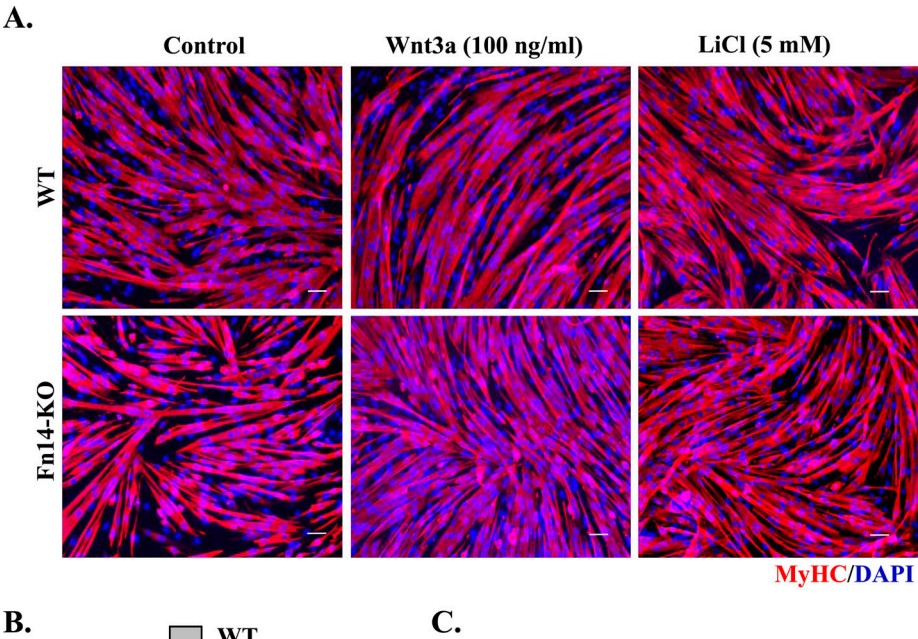

MyHC/DAPI

Figure 8. Forced activation of Wnt signaling improves myoblast fusion in Fn14-KO myoblasts.

**(A)** WT and Fn14-KO primary myoblasts were plated at equal densities and incubated in DM with a vehicle alone or 100 ng/ml Wnt3a protein or 5 mM LiCl for 48 h. The cultures were fixed and stained for MyHC (red) and DAPI (blue). Representative photomicrographs are presented here. Scale bar: 50 $\mu$m. **(B)** Quantification of the percentage of myotubes containing ≥ 10 nuclei in control-, Wnt3a-, and LiCl-treated WT and Fn14-KO cultures. n = 3 per group. **(C)** Average diameter of myotubes in control-, Wnt3a-, and LiCl-treated WT and Fn14-KO cultures. n = 3 per group. Data are presented as the mean ± SEM analyzed by a two-way ANOVA followed by Tukey's multiple-comparison test. *$P$ ≤ 0.05, values significantly different from the corresponding control myotube. #$P$ ≤ 0.05, values significantly different from corresponding WT myoblast cultures.

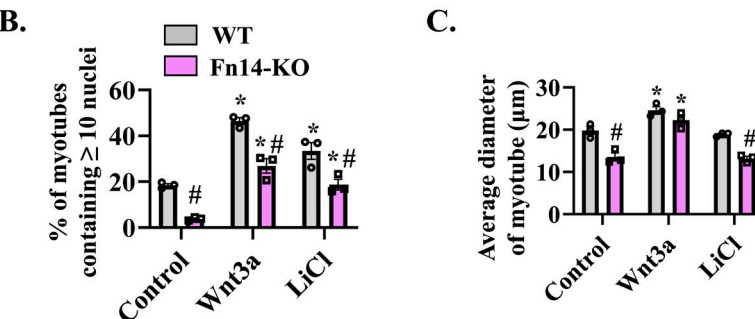

Consistent with the published report (Girgenrath et al, 2006), we found a deficit in muscle regeneration in whole-body Fn14-KO mice. Interestingly, global deletion of Fn14 in mice did not have any major effect on the expression of various MRFs or abundance of satellite cells in regenerating myofibers. It is possible that the deficiency of Fn14 is compensated by other factors present in the injured muscle microenvironment or it has no role in regulating differentiation of myogenic cells in the skeletal muscle. Skeletal muscle regeneration involves activation of multiple signaling pathways in muscle progenitor cells, regenerating myofibers, and other cell types that infiltrate the damaged muscle (Yin et al, 2013; Tidball et al, 2014; Tidball, 2017; Yang & Hu, 2018; Relaix et al, 2021). We generated mice in which Fn14 was deleted specifically in differentiated myofibers or myoblasts. Intriguingly, deletion of Fn14 in myofibers of adult mice did not affect muscle regeneration, suggesting Fn14-mediated signaling in differentiated myofibers is dispensable for skeletal muscle regeneration. In contrast, myoblast-specific deletion of Fn14 led to significant deficits in skeletal muscle regeneration (Figs 2 and 3). Interestingly, myoblast-specific deletion of Fn14 in mice augments the levels of various MRFs and abundance of satellite cells in the regenerating muscle. Although the physiological significance remained unknown, an increase in satellite cell count could be a compensatory response to counteract the deficits in muscle formation. Our experimentation revealed that the loss of

Fn14 reduces the fusion of myoblasts with injured myofibers in mice (Fig 4).

Myogenesis is a highly coordinated process that involves activation of promyogenic pathways and the expression of various MRFs (Hindi et al, 2013; Yin et al, 2013; Relaix et al, 2021). Previous studies have shown that genetic ablation or knockdown of Fn14, as well as treatment with high concentrations of TWEAK, inhibit myotube formation in cultured myoblasts following induction of differentiation (Girgenrath et al, 2006; Dogra et al, 2007a, 2007b). However, the mechanisms of action of Fn14 in myogenesis remained unknown. RNA-Seq and biochemical analyses revealed that the loss of Fn14 in myoblasts leads to the enhanced expression of muscle differentiation markers, myogenin and MyHC in normal growth conditions (Fig 5). Interestingly, although the loss of Fn14 in myoblasts leads to the formation of smaller myotubes, it does not reduce the expression of myogenin and MyHC after induction of differentiation (Fig 5G), suggesting that similar to its role in the regenerating skeletal muscle, Fn14 predominantly mediates the fusion of cultured myoblasts. The positive role of Fn14 in myoblast fusion is also evidenced by our findings that the overexpression of Fn14 in myoblasts leads to the formation of myotubes having an increased diameter following incubation in DM (Fig 5H and I). Published reports suggest that p38 MAPK plays an active role at the onset of differentiation. Indeed, p38$\alpha$/$\beta$ is involved in chromatin

remodeling and regulating transcription of muscle genes during myogenic differentiation (Perdiguero et al, 2007; Brennan et al, 2021). Furthermore, Akt1 is essential for initiation of differentiation in culture and is required for normal myoblast motility (Gardner et al, 2012). Interestingly, our results demonstrate that the overexpression of Fn14 in cultured myoblasts leads to the increased activation of both p38MAPK and Akt, which could be one of the potential mechanisms for improvements in myotube formation in Fn14-overexpressing cultures (Fig S6).

Recent studies have demonstrated that non-canonical NF-κB pathway augments the fusion of cultured myoblasts (Enwere et al, 2012; Hindi et al, 2013, 2017b). TWEAK protein activates both canonical and non-canonical NF-κB signaling pathways in different cell types, including myoblasts. Although a high concentration of TWEAK protein inhibits myogenic differentiation partly through reducing the levels of MyoD and preventing cell cycle exit (Dogra et al, 2006; Girgenrath et al, 2006), lower amounts of TWEAK induce myoblast fusion through activation of the non-canonical NF-κB signaling (Enwere et al, 2012). However, we found no difference in the levels of established markers of canonical or non-canonical NF-κB signaling in WT and Fn14-KO cultures or in regenerating myofibers of Fn14$^{fl/fl}$ and Fn14$^{myoKO}$ mice (Fig 6A and B), suggesting that Fn14-induced myoblast fusion does not involve NF-κB signaling.

Cytoplasmic Ca$^{2+}$ levels play an important role in the induction of myogenic differentiation and initiation of muscle contractions. Accumulating evidence suggests that calcium-dependent signaling also plays a critical role in myoblast fusion (Sinha et al, 2022). Elevated levels of intracellular Ca$^{2+}$ cause activation of calcineurin, which dephosphorylates the NFATc transcription factor, which allows NFATc nuclear translocation and regulation of gene transcription. Previous studies have shown that NFATc2 translocates to the nucleus after primary fusion and is essential for subsequent myonuclear accretion. Indeed, Nfatc2-null mice display myofibers with fewer nuclei and reduced regenerative capacity in vivo without affecting the number of myofibers (Horsley et al, 2001; Pavlath & Horsley, 2003). A recent study demonstrated that elevated cytosolic Ca$^{2+}$ causes activation of CaM-kinase-II (CamKII) in nascent myotubes, which is required for myocyte-to-myotube fusion. Interestingly, RyRs and SERCA2 channels that increase cytoplasmic Ca$^{2+}$ levels for muscle contractions also mediate the secondary fusion in cultured myoblasts (Eigler et al, 2021). Although exact mechanisms remain unknown, we have found that the loss of Fn14 represses the gene expression of many proteins involved in regulation of Ca$^{2+}$ homeostasis and signaling. Furthermore, the levels of NFATc2 are reduced in Fn14-KO myoblast cultures, suggesting that disruption of Ca$^{2+}$-mediated signaling may contribute to reduced fusion in cultured myoblasts (Fig 6).

Wnt signaling mediates embryonic muscle development and postnatal myogenesis (von Maltzahn et al, 2012; Hindi et al, 2013, 2017b). Indeed, recombinant Wnt1 and Wnt3a proteins augment the fusion of cultured myoblasts leading to formation of thicker myotubes (Rochat et al, 2004; Pansters et al, 2011). Similarly, activation of Wnt signaling using LiCl results in a considerable increase in the myotube diameter and the number of nuclei in each myotube (Rochat et al, 2004). During regenerative myogenesis, canonical Wnt signaling is induced in muscle progenitor cells within 2–5 d of muscle injury to augment their fusion to injured myofibers. The role

of Wnt signaling in myoblast fusion is also supported by the findings that intramuscular injection of Wnt3a protein increases the size of regenerating myofibers without changing the myofiber number (Brack et al, 2008). Our results demonstrate that the loss of Fn14 in myoblasts inhibits key components of canonical Wnt signaling both in vivo and in vitro. Importantly, treatment with Wnt3 protein or LiCl significantly improved myotube formation in Fn14-deficient cultured myoblasts (Fig 8). Furthermore, the overexpression of Fn14 leads to the increased expression of various components of Wnt signaling in cultured myoblasts (Fig 7), suggesting that Fn14 regulates activation of Wnt signaling during myogenesis and inhibition of Wnt signaling could be another potential mechanism for deficits in muscle regeneration in myoblast-specific Fn14-KO mice.

The levels of both TWEAK and Fn14 are up-regulated after muscle injury (Girgenrath et al, 2006; Mittal et al, 2010b). Although Fn14 regulates myoblast fusion during muscle regeneration in adult mice, there was no difference in the skeletal muscle mass or the myofiber CSA between WT and whole-body Fn14-KO mice (Fig 1). Similarly, the muscle mass is comparable between WT and global TWEAK-KO mice (Mittal et al, 2010a, 2010b), suggesting that both TWEAK and Fn14 are dispensable for the embryonic development of the skeletal muscle and for postnatal muscle growth. It is also possible that the role of TWEAK and Fn14 in muscle formation is compensated by other growth factors present during embryonic development and neonatal stages. The TWEAK cytokine binds to the Fn14 receptor to induce various cellular responses (Winkles, 2008; Tajrishi et al, 2014). Intriguingly, whole-body TWEAK-KO mice do not show any deficit in muscle regeneration in response to acute injury (Mittal et al, 2010b). Previously published reports have demonstrated that although low levels of TWEAK promote myoblast proliferation and fusion, overstimulation of Fn14 by TWEAK inhibits the differentiation of cultured myoblasts (Dogra et al, 2006; Enwere et al, 2012; Tajrishi et al, 2014). Similarly, the transgenic overexpression of TWEAK inhibits muscle regeneration in mice (Mittal et al, 2010b). Notably, elevated levels of TWEAK and Fn14 also cause muscle wasting, which may contribute to the delay in muscle regeneration in TWEAK-transgenic mice (Dogra et al, 2007a; Mittal et al, 2010a, 2010b). Importantly, there are reports suggesting that the Fn14 receptor can activate downstream signaling pathways even in the absence of the TWEAK cytokine (Dogra et al, 2007b; Winkles, 2008; Tajrishi et al, 2014). Indeed, our results demonstrate that the overexpression of Fn14 is sufficient to induce the fusion of cultured myoblasts (Fig 5H and I), further suggesting that Fn14-mediated signaling promotes myoblast fusion independently of the TWEAK cytokine.

In summary, our results in the present study demonstrate that Fn14 specifically regulates the myoblast fusion step during myogenic differentiation. One of the limitations of this study is that we used only one model of muscle injury. It would be interesting to determine whether Fn14 also promotes myoblast fusion during postnatal growth or during overload-induced myofiber hypertrophy. Another limitation is that we used molecular and genetic approaches to investigate the role of Fn14 in muscle regeneration after acute injury. For therapeutic purposes, it would be important to develop small molecules that can stimulate Fn14 signaling to augment myoblast fusion. Moreover, it remains unknown

whether modulating the levels of Fn14 can improve myoblast fusion and myofiber regeneration in animal models of muscle diseases, such as muscular dystrophy. Such experiments are needed before considering Fn14 as a potential molecular target for enhancing muscle regeneration and growth. Although more investigations are needed to understand the mechanisms of action of Fn14 during myogenesis, our study provides initial evidence that augmenting the levels of Fn14 in endogenous myoblasts can improve muscle regeneration and growth in degenerative muscle diseases.

# Materials and Methods

### Animals

C57BL/6 mice were purchased from Jackson Labs. Whole-body Fn14-knockout (i.e., Fn14-KO) mice have been previously described (Girgenrath et al, 2006). Floxed Fn14 (i.e., Fn14$^{fl/fl}$) mice were generated by Taconic by inserting loxP sites upstream of exon 2 and downstream of exon 4 of Fn14 as described previously (Gupta et al, 2021). Myofiber-specific Fn14-KO (henceforth Fn14$^{mKO}$) mice were generated by crossing Fn14$^{fl/fl}$ mice with MCK-Cre (strain: B6.FVB(129S4)-Tg(Ckmm-cre)5Khn/J; Jackson Laboratory) mice as described previously (Tomaz da Silva et al, 2022). Myoblast-specific Fn14-KO (Fn14$^{myoKO}$) mice were generated by crossing Fn14$^{fl/fl}$ mice with Myod1-Cre (Jax strain: FVB.Cg-Myod1 $^{tm2.1(icre)Glh}$/J) mice. All mice were in the C57BL/6 background, and their genotype was determined by PCR from tail DNA. All the experiments were performed in strict accordance with the recommendations in the Guide for the Care and Use of Laboratory Animals of the National Institutes of Health. All the animals were handled according to an approved Institutional Animal Care and Use Committee protocol (PROTO201900043) of the University of Houston. All surgeries were performed under anesthesia, and every effort was made to minimize suffering.

### Skeletal muscle injury and in vivo myoblast fusion assay

The TA muscle of adult mice was injected 50 $\mu$l of 1.2% BaCl$_2$ (Sigma Chemical Co.) dissolved in saline to induce necrotic injury as described previously (Ogura et al, 2015; Hindi and Kumar, 2016). The TA muscle was collected at different time points after injury from euthanized mice for biochemical and morphometric analysis. To study myoblast fusion in vivo, the mice were given an intraperitoneal injection of EdU (4 $\mu$g per gram body weight) at day 3 after intramuscular injection of 1.2% BaCl$_2$ into the TA muscle. On day 11 after EdU injection, the TA muscle was isolated and sectioned in a microtome cryostat. The sections were subsequently immunostained with anti-Laminin for marking boundaries of myofibers and processed for the detection of EdU$^+$ nuclei similar to as described previously (Hindi et al, 2017a). The EdU$^+$ nuclei on muscle sections were detected as instructed in the Click-iT EdU Alexa Fluor 488 Imaging Kit (Invitrogen). Finally, images were captured and the number of intramyofiber EdU$^+$ myonuclei/myofiber, percentage of 2 or more EdU$^+$ centrally nucleated fibers, and percentage of EdU$^+$ myonuclei/total nuclei were quantified

using NIH ImageJ software. To reduce variations, three to four different sections from the mid-belly of each muscle were included for analysis.

### Histology and morphometric analysis

The TA muscle was isolated from mice, snap-frozen in liquid nitrogen, and sectioned with a microtome cryostat. For the assessment of muscle morphology and quantification of the fiber CSA, 7-$\mu$m-thick transverse sections of the TA muscle were stained with H&E. The sections were examined under an Eclipse TE2000-U microscope (Nikon). For quantitative analysis, the CSA of myofibers was analyzed in H&E-stained TA muscle sections using NIH ImageJ software. For each muscle, the distribution of the fiber CSA was calculated by analyzing approximately 300 myofibers.

### Immunohistochemistry

For immunohistochemistry studies, frozen TA or plantaris muscle sections were fixed in acetone or 4% PFA in PBS, blocked in 2% BSA in PBS for 1 h, and incubated with mouse anti-Pax7, mouse anti-eMyHC, and rabbit anti-laminin, in blocking solution at 4°C overnight under humidified conditions. The sections were washed briefly with PBS before incubation with goat anti-mouse Alexa Fluor 594 and goat anti-rabbit Alexa Fluor 488 secondary antibodies for 1 h at room temperature and then washed three times for 15 min with PBS. Nuclei were counterstained with DAPI. The slides were mounted using the fluorescence medium (Vector Laboratories) and visualized at room temperature on a Nikon Eclipse TE2000-U microscope (Nikon), a digital camera (Nikon Digital Sight DS-Fi1), and NIS-Elements BR 3.00 software (Nikon). Image levels were equally adjusted using Photoshop CS6 software (Adobe).

### Myoblast isolation, culturing, and fusion assay

Primary myoblasts were prepared from the hindlimbs of 8-wk-old male or female mice following a protocol as described previously (Hindi et al, 2017a, 2017b). Briefly, hindlimb muscles were collected and minced/digested with collagenase type II in DMEM. Muscle tissues were mechanically scraped, centrifuged, and resuspended in DMEM, and cell mixture was filtered and incubated at 37°C for 72 min. After that, cells were purified by doing preplating to assure myoblast purification. To induce differentiation, the cells were incubated in a differentiation medium (DM; 2% horse serum in DMEM). Cultures were fixed with 4% PFA in PBS for 15 min at room temperature and then permeabilized with 0.1% Triton X-100 in PBS for 5–8 min. Cells were blocked with 2% BSA in PBS and incubated with mouse anti-MyHC (MF-20) overnight at 4°C and goat anti-mouse Alexa Fluor 568 at room temperature for 1 h. Nuclei were counterstained with DAPI for 3 min. Stained cells were photographed and analyzed using a fluorescent inverted microscope (Nikon Eclipse TE2000-U), a digital camera (Digital Sight DS-Fi1), and Elements BR 3.00 software (Nikon). To measure fusion efficiency, more than 100 of MyHC-positive myotubes containing 2 or more nuclei were counted. More than five field images were analyzed per experimental group. To measure the average diameter of myotubes, 100–120 myotubes per group were included. For consistency, diameters were measured at the midpoint

along with the length of the MyHC[+] myotubes. The myotube diameter was measured using NIH ImageJ software. Results presented are from four to five independent experiments. Image levels were equally adjusted using Photoshop CS6 software (Adobe).

## Generation and use of retroviruses

A pBABE-Puro empty vector and a pBABE-Puro-EGFP plasmid were purchased from Addgene. Mouse Fn14 cDNA was isolated and ligated at BamHI and SalI sites in the pBABE-Puro plasmid. The integrity of the cDNA was confirmed by performing DNA sequencing. About $5 \times 10^6$ Platinum-E packaging cells (Cell Biolabs, Inc.) were transfected with 5 $\mu$g of pBABE-Puro-Fn14 or pBABE-Puro-EGFP using FuGENE-HD (Promega). After 24 h of transfection, the medium was replaced with 10% FBS. 48 h after transfection, viral supernatants were collected, filtered through 0.45-micron filters, and then added to primary myoblasts in growth media containing 10 $\mu$g/ml polybrene. After two successive retroviral infections, cells were grown for 48 h and selected using 1.2 $\mu$g/ml puromycin for 2 wk.

## RNA isolation and qRT–PCR

RNA isolation and qRT–PCR were performed following a standard protocol as described previously. In brief, total RNA was extracted from uninjured and injured TA muscles of mice or cultured myoblasts using TRIzol reagent (Thermo Fisher Scientific) and RNeasy Mini Kit (QIAGEN) according to the manufacturers' protocols. First-strand cDNA for PCR analyses was made with a commercially available kit (iScript cDNA Synthesis Kit; Bio-Rad Laboratories). The quantification of mRNA expression was performed using the SYBR Green dye (Bio-Rad SsoAdvanced Universal SYBR Green Supermix) method on a sequence detection system (CFX384 Touch Real-Time PCR Detection System; Bio-Rad Laboratories). The sequence of the primers is described in Table S1. Data normalization was accomplished with the endogenous control ($\beta$-actin), and the normalized values were subjected to a $2^{-\Delta\Delta Ct}$ formula to calculate the fold change between control and experimental groups.

## Western blot

The relative amount of various proteins was determined by performing Western blot following a standard protocol. In brief, TA muscles of mice or cultured primary myoblasts were washed with PBS and homogenized in lysis buffer (50 mM Tris-Cl [pH 8.0], 200 mM NaCl, 50 mM NaF, 1 mM dithiothreitol, 1 mM sodium orthovanadate, 0.3% IGEPAL, and protease inhibitors). ~100 $\mu$g protein was resolved on each lane on 8–12% SDS–PAGE, transferred onto a nitrocellulose membrane, and probed using the specific primary antibody (Table S2). Bound antibodies were detected by secondary antibodies conjugated to horseradish peroxidase (Cell Signaling Technology). Signal detection was performed by an enhanced chemiluminescence detection reagent (Bio-Rad). Approximate molecular masses were determined by comparison with the migration of prestained protein standards (Bio-Rad). Uncropped gel images are presented as source data.

## RNA-sequencing and data analyses

Total RNA from WT and Fn14-KO cultures was extracted using TRIzol reagent (Thermo Fisher Scientific) using RNeasy Mini Kit (QIAGEN) according to the manufacturers' protocols. The mRNA-Seq library was prepared using poly (A)-tailed enriched mRNA at the UT Cancer Genomics Center using the KAPA mRNA HyperPrep Kit protocol (KK8581; Roche, Holding AG) and the KAPA Unique Dual-indexed Adapter kit (KK8727; Roche). The Illumina NextSeq 550 was used to produce 75 base paired-end mRNA-Seq data at an average read depth of ~38 M reads/sample. RNA-Seq fastq data were processed using CLC Genomics Workbench 20 (QIAGEN). Illumina sequencing adapters were trimmed, and reads were aligned to the mouse reference genome RefSeq GRCm39.105 from the Biomedical Genomics Analysis Plugin 20.0.1 (QIAGEN). Normalization of RNA-Seq data was performed using the trimmed mean of M-values. Genes with fold change (FC) ≥ I1.5I (or log$_2$FC ≥ I0.5I) and FDR <0.05 were assigned as differentially expressed genes and represented in a volcano plot using the ggplot function in R software (v 4.2.2). Over-representation analysis was performed with a hypergeometric test using WebGestalt (v 0.4.3) (Benjamini & Hochberg, 1995). Gene ontology (GO) biological processes associated with the up-regulated and down-regulated genes were identified with a FDR cutoff value of 0.05. Network enrichment analysis was performed using the Metascape gene annotation and analysis tool (metascape.org) as described previously (Zhou et al, 2019). Heatmaps were generated using the heatmap.2 function (Gu & Hubschmann, 2022) using z-scores calculated based on transcripts per million (TPM) values. TPM values were converted to log (TPM+1) to handle zero values. Genes involved in specific pathways were manually selected for heatmap expression plots. The mean TPM values for WT cultures in the growth medium are provided in Table S3. All the raw data files can be found on the NCBI SRA repository using the accession code PRJNA999372.

## Statistical analyses and experimental design

The sample size was calculated using power analysis methods for a priori determination based on the s.d., and the effect size was previously obtained using the experimental procedures employed in the study. For animal studies, we calculated the minimal sample size for each group as eight animals. Considering a likely drop-off effect of 10%, we set the sample size of each group of six mice. For some experiments, three to four animals were sufficient to obtain statistical significant differences. Animals of the same sex and the same age were employed to minimize physiological variability and to reduce s.d. from the mean. The exclusion criteria for animals were established in consultation with a veterinarian and experimental outcomes. In case of death, skin injury, ulceration, sickness, or weight loss of >10%, the animal was excluded from analysis. Tissue samples were excluded in cases such as freeze artifacts on histological sections or failure in extraction of RNA or protein of suitable quality and quantity. We included animals from different breeding cages by random allocation to the different experimental groups. All animal experiments were blinded using number codes until the final data analyses were performed. Statistical tests were used as described in Figure legends. Results are expressed as the

mean + SEM. Statistical analyses used a two-tailed *t* test or a two-way ANOVA followed by Tukey's multiple-comparison test to compare quantitative data populations with normal distribution and equal variance. A value of *P* ≤ 0.05 was considered statistically significant unless otherwise specified.

## Data Availability

All relevant data related to this study are available from the authors upon request. Raw data files for the RNA-Seq experiment can be found on the NCBI SRA repository using the accession code PRJNA999372.

## Supplementary Information

## Acknowledgements

We are thankful to Dr. Linda Burkly of Biogen Inc. (Cambridge, MA) for providing Fn14-knockout mice and floxed Fn14 mice. This work was supported by National Institute of Health grants AR081487 and AR059810 to A Kumar. We thank the technical support from the Cancer Prevention and Research Institute of Texas (CPRIT RP180734).

### Author Contributions

M Tomaz da Silva: conceptualization, data curation, formal analysis, investigation, and writing—original draft, review, and editing.
AS Joshi: data curation, formal analysis, and investigation.
MB Castillo: data curation and formal analysis.
TE Koike: formal analysis and investigation.
A Roy: data curation and formal analysis.
PH Gunaratne: data curation and formal analysis.
A Kumar: conceptualization, supervision, funding acquisition, project administration, and writing—review and editing.

### Conflict of Interest Statement

The authors declare that they have no conflict of interest.

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
