## [Reviewer comments · Life Science Alliance]

Life Science Alliance

Fn14 promotes myoblast fusion during regenerative myogenesis

Meiricris Tomaz da Silva, Aniket Joshi, Micah Castillo, Tatiana Koike, Anirban Roy, Preethi Guanaratne, and Ashok Kumar
DOI: <https://doi.org/10.26508/lsa.202302312>

Corresponding author(s): Ashok Kumar, University of Houston

Review Timeline:

Submission Date:	2023-08-09
Editorial Decision:	2023-09-06
Revision Received:	2023-09-19
Editorial Decision:	2023-09-22
Revision Received:	2023-09-22
Accepted:	2023-09-26

Scientific Editor: Novella Guidi

Transaction Report:

September 6, 2023

Re: Life Science Alliance manuscript #LSA-2023-02312-T

Prof. Ashok Kumar
University of Houston
Pharmacological and Pharmaceutical Sciences
4349 Martin Luther King Boulevard
Houston, TX 77204-5039

Dear Dr. Kumar,

Thank you for submitting your manuscript entitled "Fn14 promotes myoblast fusion during regenerative myogenesis" to Life Science Alliance. The manuscript was assessed by expert reviewers, whose comments are appended to this letter. We invite you to submit a revised manuscript addressing the Reviewer comments.

Thank you for this interesting contribution to Life Science Alliance. We are looking forward to receiving your revised manuscript.

Sincerely,

B. MANUSCRIPT ORGANIZATION AND FORMATTING:

Reviewer #1 (Comments to the Authors (Required)):

Myoblast fusion is essential for skeletal muscle development and regeneration. The current manuscript reported a cell-autonomous role of TNF receptor superfamily member 12A (Tnfrsf12, aka Fn14) in myoblast fusion during muscle regeneration. The authors followed the initial discovery that global deletion of the Fn14 gene from mice compromises muscle regeneration following acute injury. Such a phenotype was recapitulated in Fn14 myoblast-specific but not myofiber-specific knockout mouse models. Using the EdU pulse & chase strategy, the authors demonstrated a reduction of EdU+ myonuclear incorporation through fusion in post-injury muscle tissues of mutant mice. As a consequence, a portion of mutant muscle cells remain unfused for 14 days post-injury. The cell-autonomous effect was consolidated by gain/loss-of-function experiments performed on primary myoblasts. Interestingly, ectopic activation of Wnt signaling by ligand supplements or agonist partially rescued fusion defects in Fn14-knockout myoblasts. Together, these results demonstrate that Fn14-mediated signaling promotes myoblast fusion and skeletal muscle regeneration. Overall, the study was well designed and executed, and the results were presented in a rigorous way by adding the key information for the Method/Reagent sections and the raw images and clear labeling for Western blotting results. More importantly, this study affords a timely understanding of the key function of an inflammation-related pathway in muscle regeneration. A few minor issues were identified for consideration of revision.

1. A key advantage of transcriptome analysis over the qPCR method is the measurement of genes' absolute expression levels. Figure 5C, presumably also Figure 7A, showed Z scores which still reflect the relative nature of gene expression changes. It is thus more informative to add the mean FPKM or TPM values for the WT (GM) group in the figure legend or beside gene names in the figure itself. Such information could provide the key context and a more complete picture of the evaluations of the physiological significance and potential involvement of all Wnt ligands toward the phenotypes.

2. Tissue regeneration normally invokes the basic cellular and genetic mechanism of organogenesis. By checking the mouse development single-cell RNA-seq data (Mouse Organogenesis Cell Atlas, MOCA), Fn14 and its ligand Tweak both appeared to be expressed in muscle precursor cells. It is thus surprising that neither global nor myoblast-specific deletion of the Fn14 gene appears to affect muscle development, though the enumerations of myofiber numbers and myonucleus numbers of the myofibers would provide a more precise evaluation of myoblast fusion in the development context. It is thus encouraged to discuss the potential reasons underlying the difference between the key role of Fn14 in muscle regeneration and the dispensable role in muscle development. This also relates to the next comment.

3. It is encouraged to discuss the possibility of the Tweak-independent function of Fn14 because the knockout phenotype of the two genes appeared to be largely different or even opposite considering the previous reports of the link of Tweak with muscle atrophies and poor muscle regeneration outcomes.

4. Table S1, the sequences of Myomixer qPCR primers that detect the short (84 a.a.) and long (108 a.a.) isoforms were switched. The authors need to clarify or repeat qPCR experiments shown in Figure 4F.

Reviewer #2 (Comments to the Authors (Required)):

Summary

The current study by Tomaz da Silva and colleagues evaluated the cell-autonomous role of Fn14 in skeletal muscle during skeletal muscle regeneration. The authors showed that the genetic deletion of Fn14 in myoblasts hinders muscle regeneration. Also, the authors showed that deletion of Fn14 impairs myoblast fusion and inhibition of WNT signalling, while the activation of WNT signalling, hence fusion in myoblast. The experiments conducted in this study offer comprehensive new knowledge and advance the field. Although the manuscript is well-written, there are concerns regarding the presentation of the results and some moderate weaknesses that reduce the initial enthusiasm. A minor review is necessary to clarify certain points.

Major comments:

- None of the Western blot figures presented throughout the paper present the exact molecular weight (kDa) of the protein of interest. Please label the molecular weight in each western blot in the main and the supplemental images.
- In Figure 1F, the authors showed that eMyHC+ fibre was lower in the Fn14-KO mice compared to the WT. However, this

measurement was counted as a percentage of eMyHC+ myofibres containing two or more central nuclei. It would also be interesting to perform the analysis just of total eMyHC+ myofibres (not taking into consideration the central nuclei, as the authors present in Figure 4D) since, in the figure, S1B looks like there are no differences between WT and Fn14-KO mice comparing total eMyHC+ myofibres, beside the fact that the protein levels of eMyHC are higher in the Fn14-KO mice compared to WT.

- Please provide Fn14 protein expression (Western blot) and mRNA expression of Uninjured Fn14-KO mice compared to WT to observe the absence of protein and mRNA expression in the Fn14-KO mice before the injury.
- Figure S2: Do the Myofiber-specific FN14-KO mice present a different content of muscle stem cells (Pax7+ cells) than the FN14 fl/fl mice?
- Figure 2: Please provide TA muscle weight at 14 days post-injury from FN14fl/fl and Fn14 myoKO mice, as it was provided at 5 days post-injury.
- Figure 2C: The authors make the statement that following 5 and 14 days post-injury, an increase in mononuclear cells (related to immune cell infiltration) and extracellular matrix (ECM) deposition was observed. I recommend removing the statement saying that ECM deposition was increased. The H and E staining is not a direct measurement of ECM deposition; for that, the authors could perform a Trichrome staining and measure the actual collagen deposition. Visually, no significant increase in ECM is observed at 14 days post-injury. I would be careful with the statement.
- The rationale for collecting just 5 days post-injury is required. Why do some experiments also include 14 days post-injury (Figure 2) and others not (Figure 3)? Why leave the complete regenerative process aside at 21 days? Please explain.
- Figure 3SA and B: Rationale for the additional experiment creating the Fn14fl/wt;Myod1-Cre mice is required.
- Figure 2H: The authors say: "... is another model to study the impact of specific molecules on muscle progenitor cell function". Please provide which molecules, because the authors do not investigate any molecules in this figure, just CSA by H and E staining. Please consider to re-write this sentence.
- Figure 3: The authors in the title refer to "growth," where there is no measure of growth in Figure 3. The author measured in eMyHC expression (related to regeneration), MRF's (related to Muscle satellite cells myogenic commitment) and muscle satellite cell content. I recommend to re-write the title of the figure.
- Figure 4: The experiment in Figure 4A to E was at 14 days post-injury. Please provide the rationale of present mRNA expression from Figure 4F at 5 days post-injury and not at 14 days post-injury. Authors should present mRNA expression at 14 days post-injury, being consistent with the time-points for all Figure 4.
- Cell Culture: How were primary myoblast collected and isolated (FACS, MACS, or other)? Related to Figure 5. Please provide a brief description in the method section.
- Figure 5: The authors showed significant differences in myoblast-specific ablation of Fn14 in fusion and regeneration in previous figures; please provide a strong rationale for using the Fn14-KO primary myoblast and not the Fn14-myoKO primary myoblast for RNA-seq analysis and cell culture measurements.
- Figure 5: Please provide further analysis of the percentage of differentiation index as presented in this paper: PMID: 35848618, PMID: 23505062. This will complement the analysis presenting the number of MyHC-positive cells and provide a better understanding of the effect of Fn14 during differentiation.
- Figure 5G: Provide the "fold change graph" for Myogenin and MyHC.
- Rationale for the additional experiment creating the Fn14 overexpressing cell line is required.
- Provide the concentration of puromycin that was added for the selection and a Western blot image of the primary cells transfected in Normal Growth Media to confirm the Fn14 overexpression.

Minor comments:

- In the introduction, the authors say, "Accumulating evidence suggests that several transmembrane, cytoskeletal, and intracellular proteins mediate the fusion of myoblasts in vertebrates." After finishing this sentence, the authors must add the references that sustain their statement.
- In the introduction, the author provides this sentence: "The activity of the TWEAK-Fn14 system is enhanced because of highly induced local expression of Fn14 in injury leading to the activation of signaling pathways, such as MAPKs and canonical and non-canonical NF- κ B". Please provide the type of injury and the tissue affected to better understanding for the readers.
- Please provide the limitations and the translational relevance of the study in the discussion section.

Reviewer #3 (Comments to the Authors (Required)):

The study by Tomaz da Silva and colleagues builds on previous studies in which the role of Fn14 was evaluated in skeletal muscle regeneration using whole body KO mice. The authors were able to reproduce those results here and significantly extend them to study mice null for Fn14 in myofibers and myoblasts. The authors find that deletion of Fn14 in myoblasts, but not in myofibers, inhibits skeletal muscle regeneration in adult mice. They further show that although loss of Fn14 does not inhibit myogenic differentiation, it reduces myoblast fusion and show this both in vivo and in vitro. There are complimentary overexpression studies in which forced expression of Fn14 in cultured myoblasts leads to the formation of thicker myotubes and stimulates myoblast fusion through the activation of canonical Wnt signaling.

These are important new findings for the field that significantly extend our understanding of Fn14 biology in muscle regeneration. The experiments are logical and well interpreted, and the data strongly support each of the stated findings/conclusions. The authors are congratulated on an impressive body of work.

RESPONSE TO REVIEWERS' COMMENTS

We are highly grateful to all three reviewers for their time and efforts spent on reviewing our manuscript. We also appreciate all the suggestions provided by the reviewers. We have now addressed all the comments of the reviewers in this revised submission of our manuscript. Our pointwise response to reviewer's comments is as follows:

REVIEWER #1

Reviewer: Myoblast fusion is essential for skeletal muscle development and regeneration. The current manuscript reported a cell-autonomous role of TNF receptor superfamily member 12A (Tnfrsf12, aka Fn14) in myoblast fusion during muscle regeneration. The authors followed the initial discovery that global deletion of the Fn14 gene from mice compromises muscle regeneration following acute injury. Such a phenotype was recapitulated in Fn14 myoblast-specific but not myofiber-specific knockout mouse models. Using the EdU pulse & chase strategy, the authors demonstrated a reduction of EdU+ myonuclear incorporation through fusion in post-injury muscle tissues of mutant mice. As a consequence, a portion of mutant muscle cells remain unfused for 14 days post-injury. The cell-autonomous effect was consolidated by gain/loss-of-function experiments performed on primary myoblasts. Interestingly, ectopic activation of Wnt signaling by ligand supplements or agonist partially rescued fusion defects in Fn14-knockout myoblasts. Together, these results demonstrate that Fn14-mediated signaling promotes myoblast fusion and skeletal muscle regeneration. Overall, the study was well designed and executed, and the results were presented in a rigorous way by adding the key information for the Method/Reagent sections and the raw images and clear labeling for Western blotting results. More importantly, this study affords a timely understanding of the key function of an inflammation-related pathway in muscle regeneration. A few minor issues were identified for consideration of revision.

RESPONSE: We sincerely thank the reviewer for finding our study well designed, executed, and important for the role of inflammatory pathways in muscle regeneration.

Reviewer Comment # 1: A key advantage of transcriptome analysis over the qPCR method is the measurement of genes' absolute expression levels. Figure 5C, presumably also Figure 7A, showed Z scores which still reflect the relative nature of gene expression changes. It is thus more informative to add the mean FPKM or TPM values for the WT (GM) group in the figure legend or beside gene names in the figure itself. Such information could provide the key context and a more complete picture of the evaluations of the physiological significance and potential involvement of all Wnt ligands toward the phenotypes.

RESPONSE: We have taken into consideration this suggestion to include the mean of TPM values for the WT (GM) group for Figure 5C and Figure 7A in our analysis. We have incorporated a table containing these data in the supplemental data file. Please refer to new supplemental **Table S3**.

Reviewer Comment # 2: Tissue regeneration normally invokes the basic cellular and genetic mechanism of organogenesis. By checking the mouse development single-cell RNA-Seq data (Mouse Organogenesis Cell Atlas, MOCA), Fn14 and its ligand, Tweak both appeared to be expressed in muscle precursor cells. It is thus surprising that neither global nor myoblast-specific deletion of the Fn14 gene appears to affect muscle development, though the enumerations of myofiber numbers and myonucleus numbers of the myofibers would provide a more precise evaluation of myoblast fusion in the development context. It is thus encouraged to discuss the potential reasons underlying the difference between the key role of

Fn14 in muscle regeneration and the dispensable role in muscle development. This also relates to the next comment.

RESPONSE: This is a good point. Indeed, the expression of both TWEAK and Fn14 goes up after muscle injury. We have now included the discussion about this aspect in the Discussion section of our manuscript (Page # 16 and 17, highlighted text).

Reviewer Comment # 3. It is encouraged to discuss the possibility of the Tweak-independent function of Fn14 because the knockout phenotype of the two genes appeared to be largely different or even opposite considering the previous reports of the link of Tweak with muscle atrophies and poor muscle regeneration outcomes.

RESPONSE: This is a very good suggestion. Indeed, there are many reports suggesting that Fn14 can activate downstream signaling independent of TWEAK cytokine. Even in this manuscript, our results demonstrate that overexpression of Fn14 in myoblasts is sufficient to augment myotube formation (Fig. 5). We have now discussed this issue in the Discussion section of this revised manuscript (Page # 16, 17, highlighted text).

Reviewer Comment # 4, Table S1, the sequences of Myomixer qPCR primers that detect the short (84 aa) and long (108 aa) isoforms were switched. The authors need to clarify or repeat qPCR experiments shown in Figure 4F.

RESPONSE: We appreciate your observation and bringing this to our attention. We have now made this correction in the Results section and in Figure 4F. The primer names are corrected in Supplemental Table S1.

REVIEWER #2

Reviewer: The current study by Tomaz da Silva and colleagues evaluated the cell-autonomous role of Fn14 in skeletal muscle during skeletal muscle regeneration. The authors showed that the genetic deletion of Fn14 in myoblasts hinders muscle regeneration. Also, the authors showed that deletion of Fn14 impairs myoblast fusion and inhibition of WNT signalling, while the activation of WNT signalling, hence fusion in myoblast. The experiments conducted in this study offer comprehensive new knowledge and advance the field. Although the manuscript is well-written, there are concerns regarding the presentation of the results and some moderate weaknesses that reduce the initial enthusiasm. A minor review is necessary to clarify certain points.

RESPONSE: We thank the reviewer for valuable feedback on our manuscript.

Major comments:

Reviewer Comment # 1: None of the Western blot figures presented throughout the paper present the exact molecular weight (kDa) of the protein of interest. Please label the molecular weight in each western blot in the main and the supplemental images.

RESPONSE: We have now added molecular weight of all the protein in immunoblots presented in the manuscript.

Reviewer Comment # 2: In Figure 1F, the authors showed that eMyHC⁺ fibre was lower in the Fn14-KO mice compared to the WT. However, this measurement was counted as a percentage of eMyHC⁺ myofibres containing two or more central nuclei. It would also be interesting to perform the analysis just of total eMyHC⁺ myofibres (not taking into consideration the central nuclei, as the authors present in Figure 4D) since, in the figure, S1B looks like there are no differences between WT and Fn14-KO mice comparing total eMyHC⁺

myofibres, beside the fact that the protein levels of eMyHC are higher in the Fn14-KO mice compared to WT.

RESPONSE: Based on reviewer's suggestion, we have now quantified total number of eMyHC⁺ myofibers as well. However, there was no significant differences in the total number of eMyHC⁺ myofibers between the WT and Fn14-KO mice. These results are now included as Supplemental Figure S1C.

Reviewer Comment # 3: Please provide Fn14 protein expression (Western blot) and mRNA expression of uninjured Fn14-KO mice compared to WT to observe the absence of protein and mRNA expression in the Fn14-KO mice before the injury.

RESPONSE: We would like to emphasize that Fn14 protein levels are almost undetectable in adult skeletal muscle of mice. Fn14 is a highly inducible gene. We have published many articles where we found that Fn14 levels are induced in the condition of muscle injury or denervation-induced muscle atrophy (PubMed ID 20308426, 20724600). In a recent publication from our lab, we showed that Fn14 protein was undetectable in normal muscle but was drastically induced upon denervation. However, this increase was absent in muscle specific Fn14-KO mice (Tomaz Da Silva et al. FASEB J. 2022 Dec;36(12):e22666). Reviewer will also find from our experiments in this manuscript that Fn14 protein is almost absent in uninjured muscle of adult mice but drastically upregulated after injury (Figure 1B). For these reasons, we did not include the levels of Fn14 protein in uninjured muscle. However, our results with cultured myoblasts confirm complete absence of Fn14 protein in myoblasts prepared from Fn14-KO mice (Figure 5G).

Reviewer Comment # 4: Figure S2: Do the Myofiber-specific FN14-KO mice present a different content of muscle stem cells (Pax7⁺ cells) than the FN14fl/fl mice?

RESPONSE: We performed Pax7 staining on whole body Fn14-KO mice and myoblast-specific Fn14-KO mice because these mice showed deficit in muscle regeneration. However, myofiber-specific Fn14-KO mice had no muscle regeneration phenotype (H&E staining and eMyHC staining and their quantification in Figure S2) and therefore we did not perform any further analysis, including Pax7 staining.

Reviewer Comment # 5: Figure 2: Please provide TA muscle weight at 14 days post-injury from FN14fl/fl and Fn14myoKO mice, as it was provided at 5 days post-injury.

RESPONSE: This has now been included as Figure 2C of the revised manuscript.

Reviewer Comment # 6: Figure 2C: The authors make the statement that following 5 and 14 days post-injury, an increase in mononuclear cells (related to immune cell infiltration) and extracellular matrix (ECM) deposition was observed. I recommend removing the statement saying that ECM deposition was increased. The H and E staining is not a direct measurement of ECM deposition; for that, the authors could perform a Trichrome staining and measure the actual collagen deposition. Visually, no significant increase in ECM is observed at 14 days post-injury. I would be careful with the statement.

RESPONSE: We sincerely appreciate reviewer's suggestion. We agree that there is no visible ECM deposition in H&E stained sections. We have now removed ECM part from these results.

Reviewer Comment # 7: The rationale for collecting just 5 days post-injury is required. Why do some experiments also include 14 days post-injury (Figure 2) and others not (Figure 3)? Why leave the complete regenerative process aside at 21 days? Please explain.

RESPONSE: At 5 days post-injury, we can observe the initial phases of muscle regeneration, including inflammation, immune cell infiltration, and activation of satellite cells (muscle stem cells). This time point allows for the assessment of early changes in gene expression and protein production that are critical for initiating the repair process. By 14 days, we can observe advanced stages of muscle regeneration, including the formation of new muscle fibers and the resolution of inflammation. Muscle fibers may start to regain their structure, and the tissue may show signs of functional recovery. Studying muscle tissue at this stage provides insights into the progression of healing, tissue remodeling, and the potential long-term effects of the injury or any possible delay in skeletal muscle regeneration. Analyzing muscle regeneration at both 5 days and 14 days post-injury provides a more comprehensive view of the entire healing process, from the initial response to later stages of tissue repair and remodeling. Our results confirm that regeneration deficit persist on day 14 post-injury. Complete muscle regeneration occurs by day 21 in wild-type mice even though regenerated myofibers have centrally located nuclei. We could have included H&E staining at day 21 but this experiment would not provide additional insight about role of Fn14 in muscle regeneration.

Reviewer Comment # 8: Figure 3SA and B: Rationale for the additional experiment creating the Fn14^{fl/wt};Myod1-Cre mice is required.

RESPONSE: This was a control experiment to confirm that the phenotype is due to deletion of Fn14, not because of loss of MyoD from one allele. Myod1-Cre is a knock-in mouse in which Cre has been inserted at one of the allele of Myod1 gene. Since Myod1 is a critical regulator of myogenesis, we just wanted to confirm that the lack of one allele of Myod1 is not the reason for the observed phenotype in myoblast-specific Fn14-KO mice. The rationale for this experiment has been added in the Results section (Page # 7, highlighted text).

Reviewer Comment # 9: Figure 2H: The authors say: "..., is another model to study the impact of specific molecules on muscle progenitor cell function". Please provide which molecules, because the authors do not investigate any molecules in this figure, just CSA by H and E staining. Please consider to re-write this sentence.

RESPONSE: This was an inaccurate statement. We did not mean any specific molecules here. We have now modified this sentence. The new sentence is "Two consecutive injuries carried out three or four weeks apart, with enough time elapse allowing the regeneration of the muscle after first injury is another model to study muscle progenitor cell function, maintenance, or depletion" (Page # 7, last paragraph, highlighted text).

Reviewer Comment # 10: Figure 3: The authors in the title refer to "growth," where there is no measure of growth in Figure 3. The author measured in eMyHC expression (related to regeneration), MRF's (related to Muscle satellite cells myogenic commitment) and muscle satellite cell content. I recommend to re-write the title of the figure.

RESPONSE: We completely agree with the reviewer. We have now revised the title for the description of this figure. The new title is "Myoblast-specific ablation of Fn14 reduces myofiber formation" which more accurately reflect the content of the figure (Page # 8, highlighted title).

Reviewer Comment # 11: Figure 4: The experiment in Figure 4A to E was at 14 days post-injury. Please provide the rationale of present mRNA expression from Figure 4F at 5 days post-injury and not at 14 days post-injury. Authors should present mRNA expression at 14 days post-injury, being consistent with the time-points for all Figure 4.

RESPONSE: The choice of analyzing mRNA expression at 5 days post-injury in Figure 4F was driven by the dynamics of muscle regeneration. It is well documented in literature that myoblast fusion exhibit a peak between 4-7 days after injury followed by maturation and functional recovery phase (e.g. Forcina et al. Mechanisms Regulating Muscle Regeneration: Insights into the Interrelated and Time-Dependent Phases of Tissue Healing. *Cells*. 2020; 9(5):1297. <https://doi.org/10.3390/cells9051297>). Therefore, we performed this experiment at day 5 post-injury to accurately capture changes in the regulators of myoblast fusion in vivo. We have now added the rationale in the Results section of the manuscript (Page # 9, highlighted text).

Reviewer Comment # 12: Cell Culture: How were primary myoblast collected and isolated (FACS, MACS, or other)? Related to Figure 5. Please provide a brief description in the method section.

RESPONSE: We have a standard protocol in our lab to isolate primary myoblasts from skeletal muscle of adult mice. The detailed protocol has been described in one of our publications (Hindi et al. *Bio Protoc*. 2017 May 5;7(9):e2248). This is cited as Reference # 60 in this manuscript. However, we have also now added a brief description of this protocol in our manuscript (Page # 19, highlighted text).

Reviewer Comment # 13: Figure 5: The authors showed significant differences in myoblast-specific ablation of Fn14 in fusion and regeneration in previous figures; please provide a strong rationale for using the Fn14-KO primary myoblast and not the Fn14-myoKO primary myoblast for RNA-Seq analysis and cell culture measurements.

RESPONSE: We find deficits in muscle regeneration in both global Fn14-KO mice and myoblast-specific Fn14-KO mice. We have used primary myoblasts from whole body Fn14-KO mice because that would eliminate any minor effect that may occur due to deletion of one allele of *Myod1* in Fn14^{myoKO} mice. We have added a sentence in the Results section as rationale for using whole body Fn14-KO myoblasts (Page # 9, last paragraph, highlighted text)

Reviewer Comment # 14: Figure 5: Please provide further analysis of the percentage of differentiation index as presented in this paper: PMID: 35848618, PMID: 23505062. This will complement the analysis presenting the number of MyHC-positive cells and provide a better understanding of the effect of Fn14 during differentiation.

RESPONSE: We appreciate this valuable feedback. We have now performed additional analysis and presented fusion index as suggested by the reviewer. Please refer to Figure 5F.

Reviewer Comment # 15: Figure 5G: Provide the "fold change graph" for Myogenin and MyHC.

RESPONSE: This was one of the two representative experiments (not in triplicate). The fold change graph for Myogenin and MyHC have now been included in this revised submission (supplemental Figure S5).

Reviewer Comment # 16: Rationale for the additional experiment creating the Fn14 overexpressing cell line is required.

RESPONSE: Since deletion of Fn14 inhibits myotube formation, we wanted to investigate whether forced expression of Fn14 would improve myotube formation. This is a complementary approach to study the role of Fn14 in myoblasts. Moreover, there are reports suggesting that Fn14 receptor can function independent of its ligand TWEAK. The results of this experiment demonstrate that overexpression of Fn14 alone augments myotube formation

after induction of myogenic differentiation further supporting TWEAK independent role of Fn14 in myogenesis. We have now added the rationale for this experiment in the Result section (Page # 10, highlighted text)

Reviewer Comment # 17. Provide the concentration of puromycin that was added for the selection and a Western blot image of the primary cells transfected in Normal Growth Media to confirm the Fn14 overexpression.

RESPONSE: We used 1.2 μ g/ml puromycin. This has been now added in the Materials and Methods section of the revised manuscript (Page #20, highlighted text). We had performed Western blot to confirm Fn14 overexpression in transfected cells in Growth Medium as well as in Differentiation Medium. Please refer to supplemental Figure S6A, B.

Minor comments

Reviewer Comment # 18. In the introduction, the authors say, "Accumulating evidence suggests that several transmembrane, cytoskeletal, and intracellular proteins mediate the fusion of myoblasts in vertebrates." After finishing this sentence, the authors must add the references that sustain their statement.

RESPONSE: We have now added references to support this statement.

Reviewer Comment # 19: In the introduction, the author provides this sentence: "The activity of the TWEAK-Fn14 system is enhanced because of highly induced local expression of Fn14 in injury leading to the activation of signaling pathways, such as MAPKs and canonical and non-canonical NF- κ B". Please provide the type of injury and the tissue affected to better understanding for the readers.

RESPONSE: The levels of Fn14 have been found to upregulated in many tissues following injury, including skeletal muscle, arteries, liver, and in various diseases including multiple sclerosis; RA, rheumatoid arthritis; SLE, systemic lupus erythematosus, and cancer. Instead of citing individual articles, we have cited a few review articles, which discuss all these studies. We have now also mentioned a few tissues in this sentence (Page # 3, highlighted text)

Reviewer Comment # 20: Please provide the limitations and the translational relevance of the study in the discussion section.

RESPONSE: We have now added a few limitations and translational aspect of this study in the "Discussion: section of this revised manuscript (Page # 17, highlighted text).

REVIEWER #3

The study by Tomaz da Silva and colleagues builds on previous studies in which the role of Fn14 was evaluated in skeletal muscle regeneration using whole body KO mice. The authors were able to reproduce those results here and significantly extend them to study mice null for Fn14 in myofibers and myoblasts. The authors find that deletion of Fn14 in myoblasts, but not in myofibers, inhibits skeletal muscle regeneration in adult mice. They further show that although loss of Fn14 does not inhibit myogenic differentiation, it reduces myoblast fusion and show this both in vivo and in vitro. There are complimentary overexpression studies in which forced expression of Fn14 in cultured myoblasts leads to the formation of thicker myotubes and stimulates myoblast fusion through the activation of canonical Wnt signaling. These are important new findings for the field that significantly extend our understanding of Fn14 biology in muscle regeneration. The experiments are logical and well interpreted, and the data strongly support each of the stated findings/conclusions. The authors are congratulated on an impressive body of work.

RESPONSE: We are grateful for reviewer's encouraging words and acknowledgment of our efforts. Thank you.

September 22, 2023

RE: Life Science Alliance Manuscript #LSA-2023-02312-TR

Prof. Ashok Kumar
University of Houston
Pharmacological and Pharmaceutical Sciences
4349 Martin Luther King Boulevard
Houston, TX 77204-5039

Dear Dr. Kumar,

Thank you for submitting your revised manuscript entitled "Fn14 promotes myoblast fusion during regenerative myogenesis". We would be happy to publish your paper in Life Science Alliance pending final revisions necessary to meet our formatting guidelines.

- please add your main and supplementary figure legends to the main manuscript text after the references section
- please use the [10 author names et al.] format in your references (i.e., limit the author names to the first 10)
- we encourage you to revise the figure legends for figures 7 and S6 such that the figure panels are introduced in an alphabetical order
- please add a callout for Figure S6E to your main manuscript text
- any uploaded data should be mentioned in the Data Availability statement with accession codes

Figure Checks:

-Figure S7 is source data. This should be uploaded as Source Data instead of a Supplemental Figure, with each Source Data file relating to one figure.

A. FINAL FILES:

B. MANUSCRIPT ORGANIZATION AND FORMATTING:

Sincerely,

September 26, 2023

RE: Life Science Alliance Manuscript #LSA-2023-02312-TRR

Prof. Ashok Kumar
University of Houston
Pharmacological and Pharmaceutical Sciences
4349 Martin Luther King Boulevard
Houston, TX 77204-5039

Dear Dr. Kumar,

Thank you for submitting your Research Article entitled "Fn14 promotes myoblast fusion during regenerative myogenesis". It is a pleasure to let you know that your manuscript is now accepted for publication in Life Science Alliance. Congratulations on this interesting work.

DISTRIBUTION OF MATERIALS:

Again, congratulations on a very nice paper. I hope you found the review process to be constructive and are pleased with how the manuscript was handled editorially. We look forward to future exciting submissions from your lab.

Sincerely,
